# Health education for patients with acute coronary syndrome and type 2 diabetes mellitus: an umbrella review of systematic reviews and meta-analyses

Xian-liang Liu,[1,2,3] Yan Shi,[1] Karen Willis,[4] Chiung-Jung (Jo) Wu,[5,6,7,8] Maree Johnson[9,10]

For numbered affiliations see end of article.

**Correspondence to**
Xian-liang Liu;
liu.xianliang@myacu.edu.au

## ABSTRACT

**Objectives** This umbrella review aimed to identify the current evidence on health education-related interventions for patients with acute coronary syndrome (ACS) or type two diabetes mellitus (T2DM); identify the educational content, delivery methods, intensity, duration and setting required. The purpose was to provide recommendations for educational interventions for high-risk patients with both ACS and T2DM.

**Design** Umbrella review of systematic reviews and meta-analyses.

**Setting** Inpatient and postdischarge settings.

**Participants** Patients with ACS and T2DM.

**Data sources** CINAHL, Cochrane Library, Joanna Briggs Institute, Journals@Ovid, EMBase, Medline, PubMed and Web of Science databases from January 2000 through May 2016.

**Outcomes measures** Clinical outcomes (such as glycated haemoglobin), behavioural outcomes (such as smoking), psychosocial outcomes (such as anxiety) and medical service use.

**Results** Fifty-one eligible reviews (15 for ACS and 36 for T2DM) consisting of 1324 relevant studies involving 2 88 057 patients (15 papers did not provide the total sample); 30 (58.8%) reviews were rated as high quality. Nurses only and multidisciplinary teams were the most frequent professionals to provide education, and most educational interventions were delivered postdischarge. Face-to-face sessions were the most common delivery formats, and many education sessions were also delivered by telephone or via web contact. The frequency of educational sessions was weekly or monthly, and an average of 3.7 topics was covered per education session. Psychoeducational interventions were generally effective at reducing smoking and admissions for patients with ACS. Culturally appropriate health education, self-management educational interventions, group medical visits and psychoeducational interventions were generally effective for patients with T2DM.

**Conclusions** Results indicate that there is a body of current evidence about the efficacy of health education, its content and delivery methods for patients with ACS or T2DM. These results provide recommendations about the content for, and approach to, health education intervention for these high-risk patients.

### Strengths and limitations of this study

► This umbrella review is the first synthesis of systematic reviews or meta-analyses to consider health education-related interventions for patients with acute coronary syndrome (ACS) or type two diabetes mellitus (T2DM).
► These results provide recommendations about the content of a health education intervention for patients with ACS and T2DM.
► The diversity of the educational interventions seen in the reviews included in this umbrella review may reflect the uncertainty about the optimal strategy for providing health education to patients.
► This umbrella review found no reviews focused on patients with ACS and T2DM—the intended target group; instead, all of the systematic reviews and meta-analyses focused on only one of these two diseases.

## INTRODUCTION

Acute coronary syndrome (ACS) is the leading cause of death worldwide. The risk of high mortality rates relating to ACS is markedly increased after an initial cardiac ischaemic event.[1] Globally, 7.2 million (13%) deaths are caused by coronary artery disease (CAD),[2] and it is estimated that >7 80 000 persons will experience ACS each year in the USA.[3] Moreover, about 20%–25% of patients with ACS reportedly also have diabetes mellitus (DM); predominantly type two diabetes mellitus (T2DM).[4 5] Patients with ACS and DM have an increased risk of adverse outcomes such as death, recurrent myocardial infarction (MI), readmission or heart failure during follow-up.[6] Longer median delay times from symptom onset to hospital presentation, have been reported among patients with ACS and DM than patients with ACS alone.[7]

DM is now considered to confer a risk equivalent to that of CAD for patients for future MI and cardiovascular mortality.[8] Mortality

was significantly higher among patients with ACS and DM than among patients with ACS only following either ST segment elevation myocardial infarction (STEMI) (8.5% (ACS and DM) vs 5.4% (ACS)) or unstable angina/non-STEMI (NSTEMI) (2.1% (ACS and DM) vs 1.1% (ACS)).[9] ACS and T2DM are often associated with high-risk factors such as low levels of physical exercise, obesity, smoking and unhealthy diet.[10] Some of these and other risk factors, specifically glycaemia, high blood pressure (BP), lipidaemia and obesity, are frequently addressed by health education interventions.[10]

Health education interventions are comprehensive programmes that healthcare providers deliver to patients aimed at improving patients' clinical outcomes through the increase and maintenance of health behaviours.[11] Along with education about, for example, medication taking, these programmes seek to increase behaviours such as physical exercise and a healthy diet thus reducing patient morbidity or mortality.[11] Most diabetes education is provided through programmes within outpatient services or physicians' practices.[12] Many recent education programmes have been designed to meet national or international education standards[13–15] with diabetes education being individualised to consider patients' existing needs and health conditions.[16] Patients with T2DM have reported feelings of hopelessness and fatigue with low levels of self-efficacy, after experiencing an acute coronary episode.[17]

Although there are numerous systematic reviews of educational interventions relating to ACS or T2DM, an umbrella review providing direction on educational interventions for high-risk patients with both ACS and T2DM is not available, indicating a need to gather the current evidence and develop an optimal protocol for health education programmes for patients with ACS and T2DM. This umbrella review will examine the best available evidence on health education-related interventions for patients with ACS or T2DM. We will synthesise these findings to provide direction for health education-related interventions for high-risk patients with both ACS and T2DM.

An umbrella review is a new method to summarise and synthesise the evidence from multiple systematic reviews/meta-analyses into one accessible publication.[18] Our aim is to systematically gather, evaluate and organise the current evidence relating the health education interventions for patients with ACS or T2DM, and proffer recommendations for the scope of educational content and delivery methods that would be suitable for patients with ACS and T2DM.

## METHODS
### Data sources
This umbrella review performed a literature search to identify systematic reviews and meta-analyses examining health education-related interventions for patients with ACS or T2DM. The search strategies are described in online supplementary appendix 1. This umbrella review searched eight databases for articles published from January 2000 to May 2016: CINAHL, Cochrane Library, Joanna Briggs Institute, Journals@Ovid, EMBase, Medline, PubMed and Web of Science. The search was limited to English language only. The following broad MeSH terms were used: *acute coronary syndrome; angina, unstable; angina pectoris; coronary artery disease; coronary artery bypass; myocardial infarction; diabetes mellitus, type two; counseling; health education; patient education as topic; meta-analysis (publication type);* and *meta-analysis* as a topic.

### Inclusion criteria
#### Participants
All participants were diagnosed with ACS or T2DM using valid, established diagnostic criteria. The diagnostic standards included those described by the American College of Cardiology or American Heart Association,[3] National Heart Foundation of Australia and Cardiac Society of Australia and New Zealand,[19] WHO[20] or other associations.

#### Intervention types
For this umbrella review, health education-related interventions refer to any planned activities or programmes that include behaviour modification, counselling and teaching interventions. Results considered for this review included changes in clinical outcomes (including BP levels, body weight, diabetes complications, glycated haemoglobin (HbA1c), lipid levels, mortality rate and physical activity levels), behavioural outcomes (such as diet, knowledge, self-management skills, self-efficacy and smoking), psychosocial outcomes (such as anxiety, depression, quality of life and stress) and medical service use (such as medication use, healthcare utilisation and cost-effectiveness) for patients with ACS or T2DM. These activities or programmes included any educational interventions delivered to patients with ACS or T2DM. The interventions are delivered in any format, including face-to-face, telephone and group-based or one-on-one, and the settings include community, hospital and home. The interventions were delivered by nurses (including diabetes nurse educators), physicians, community healthcare workers, dietitians, lay people, rehabilitation therapists or multidisciplinary teams.

#### Study types
Only systematic reviews and meta-analyses were included in this review.

#### Eligibility assessment
The title and abstract of all of the retrieved articles were assessed independently by two reviewers (XL-L, YS) based on the inclusion criteria. All duplicate articles were identified within EndNote V.X7[21] and subsequently excluded. If the information from the titles and abstract was not clear, the full articles were retrieved. The decision to include an article was based on an appraisal of the full text of all retrieved articles. Any disagreements during this process

were settled by discussion and, if necessary, consensus was sought with a third reviewer. We developed an assessment form in which specific reasons for exclusion were detailed.

## Assessment of methodological quality

The methodological quality and risk of bias were assessed for each of the included publications using the Assessment of Multiple Systematic Reviews (AMSTAR),[22] independently by the same two reviewers (see table 1). The AMSTAR is an 11-item tool, with each item provided a score of 1 (specific criterion is met) or 0 (specific criterion is not met, unclear or not applicable).[22 23] An overall score for the review methodological quality is then calculated as the sum of the individual item scores: high quality, 8–11; medium quality, 4–7 or low quality, 0–3.[23] If the required data were not available in the article, the original authors were contacted for more information. The low quality reviews (AMSTAR scale: 0–3) were excluded in this umbrella review.

## Data extraction

Data were independently extracted by two reviewers using a predefined data extraction form. For missing or unclear information, the primary authors were contacted for clarification.

## Statistical presentation of results from reviews

All of the results were extracted for each included systematic review or meta-analysis, and the overall effect estimates are presented in a tabular form. The number of systematic reviews or meta-analyses that reported the outcome, total sample (from included publications) and information of health education interventions is also presented in tables 2 and 3.[24] A final 'summary of evidence' was developed to present the intervention, included study synthesis, and indication of the findings from the included papers (table 4).[24] This umbrella review calculated the corrected covered area (CCA) (see online supplementary appendices 2 and 3). The CCA statistic is a measure of overlap of trials (the repeated inclusion of the same trial in subsequent systematic reviews included in an umbrella systematic review). A detailed description of the calculation is provided by the authors who note slight CCA as 0%–5%, moderate CCA as 6%–10%, high CCA as 11%–15% and very high CCA is >15%.[25] The lower the CCA the lower the likelihood of overlap of trials included in the umbrella review.

## Synthesising the results and rating the evidence for effectiveness

The statements of evidence were based on a rating scheme to gather and rate the evidence across the included publications.[26] The statements of evidence were based on the following rating scheme: *sufficient evidence*, sufficient data to support decisions about the effect of the health education-related interventions.[26] A rating of *sufficient evidence* in this review is obtained when systematic reviews or meta-analyses with a large number of included articles

or participants produce a statistically significant result between the health education group and the control group.[26] *Some evidence*, is a less conclusive finding about the effects of the health education-related interventions[26] with statistically significant findings found in only a few included reviews or studies. *Insufficient evidence*, refers to not enough evidence to make decisions about the effects of the health education-related interventions, such as non-significant results between the health education group and the control group in the included systematic reviews or meta-analyses.[26] *Insufficient evidence to determine*, refers to not enough pooled data to be able to determine whether of the health education-related interventions are effective or not based on the included reviews.[26]

# RESULTS

## Characteristics of included reviews

The selection process and number of studies at each step was illustrated as presented in figure 1. The database search yielded 692 publications, with removal of 197 duplicates and 371 articles that did not meet the inclusion criteria, 124 full-text articles were retrieved after applying the methodological quality rating (AMSTAR scale),[27–29] and three studies[27–29] were removed due to low scores ≤3 on the AMSTAR scale. Fifty-one systematic reviews or meta-analyses[30–80] conducted between 2001 and 2016 and published in English were included (figure 1; tables 1–3); 15 relating to ACS. The overlap of the trials included in the 15 reviews and meta-analyses related to ACS was slight (CCA=2.6%). For the 36 systematic reviews relating to T2DM, the overlap of trials within these 35 reviews and meta-analyses (one review[47] did not report the included studies) was slight (CCA=2.1%). None of the articles included patients with both ACS and T2DM. The umbrella review involved a total of 277 493 patients, including 225 034 patients with coronary heart disease or ACS (one article did not report the total sample) and 52 459 patients with T2DM (16 papers did not report the total sample). The average sample size of included articles was 8161 (range, 536–68 556) participants, however, 63 studies related to ACS and 177 studies related to T2DM were included in more than one systematic review or meta-analysis (see online supplementary appendices 2 and 3 and CCA statistics). The sample of these studies would therefore be included more than once. Of the included systematic reviews or meta-analyses, 11 were published in *The Cochrane Library*. Nine of the articles described meta-analyses, 29 articles described systematic reviews and the remaining 13 articles were described as systematic reviews and meta-analyses or meta-regressions or narrative reviews.

Electronic database searches were conducted for all systematic reviews or meta-analyses, with an average of 6 databases searched (range, 2–16). The dates searched ranged widely from inception of the database through December 2014. Most of the included reviews were randomised controlled trials (RCTs), and an average of

**Table 1** Methodological quality assessment of included systematic reviews and meta-analyses

| | Systematic review/meta-analysis | Item 1 | Item 2 | Item 3 | Item 4 | Item 5 | Item 6 | Item 7 | Item 8 | Item 9 | Item 10 | Item 11 | Total score |
|---|---|---|---|---|---|---|---|---|---|---|---|---|---|
| **Systematic reviews and meta-analysis involved patients with ACS** | | | | | | | | | | | | | |
| 1 | Barth et al[69] | Yes | Yes | Yes | Yes | Yes | Yes | Yes | Yes | Yes | Yes | Yes | 11 |
| 2 | Devi et al[44] | Yes | Yes | Yes | Yes | Yes | Yes | Yes | Yes | Yes | NA | Yes | 10 |
| 3 | Ghisi et al[50] | CA | Yes | Yes | Yes | No | Yes | Yes | Yes | Yes | No | No | 7 |
| 4 | Kotb et al[59] | CA | Yes | Yes | Yes | No | Yes | Yes | Yes | Yes | No | Yes | 8 |
| 5 | Brown et al[37] | Yes | No | Yes | CA | No | Yes | Yes | Yes | Yes | NA | Yes | 7 |
| 6 | Dickens et al[45] | CA | Yes | Yes | CA | No | Yes | Yes | Yes | Yes | Yes | Yes | 8 |
| 7 | Aldcroft et al[31] | CA | No | Yes | CA | NO | Yes | Yes | Yes | Yes | No | Yes | 6 |
| 8 | Brown et al[70] | Yes | Yes | Yes | Yes | Yes | Yes | Yes | Yes | Yes | NA | Yes | 10 |
| 9 | Huttunen-Lenz et al[56] | CA | No | Yes | CA | No | Yes | Yes | Yes | Yes | No | No | 5 |
| 10 | Goulding et al[51] | Yes | Yes | Yes | CA | No | Yes | Yes | Yes | Yes | No | Yes | 8 |
| 11 | Auer et al[34] | CA | Yes | Yes | CA | No | No | Yes | No | Yes | Yes | No | 5 |
| 12 | Barth et al[36] | Yes | Yes | Yes | No | Yes | Yes | Yes | Yes | Yes | Yes | Yes | 10 |
| 13 | Fernandez et al[48] | Yes | Yes | Yes | Yes | Yes | Yes | Yes | Yes | Yes | No | No | 8 |
| 14 | Barth et al[35] | CA | Yes | Yes | CA | No | Yes | CA | Yes | Yes | Yes | Yes | 7 |
| 15 | Clark et al[41] | CA | Yes | Yes | CA | No | Yes | Yes | Yes | Yes | Yes | Yes | 8 |
| **Systematic reviews and meta-analysis involved patients with T2DM** | | | | | | | | | | | | | |
| 16 | Choi et al[40] | CA | Yes | Yes | No | No | Yes | Yes | Yes | Yes | Yes | Yes | 8 |
| 17 | Creamer et al[42] | Yes | Yes | Yes | CA | No | Yes | Yes | Yes | Yes | No | Yes | 8 |
| 18 | Huang et al[55] | CA | CA | Yes | CA | No | Yes | Yes | Yes | Yes | Yes | Yes | 7 |
| 19 | Chen et al[39] | CA | CA | Yes | CA | No | Yes | Yes | Yes | Yes | Yes | Yes | 7 |
| 20 | Pillay et al[71] | Yes | No | Yes | Yes | No | Yes | Yes | Yes | Yes | Yes | Yes | 9 |
| 21 | Terranova et al[72] | CA | CA | Yes | No | Yes | Yes | Yes | Yes | Yes | Yes | Yes | 8 |
| 22 | Attridge et al[33] | Yes | Yes | Yes | Yes | Yes | Yes | Yes | Yes | Yes | No | Yes | 10 |
| 23 | Odnoletkova et al[66] | Yes | CA | Yes | CA | No | No | Yes | Yes | Yes | Yes | No | 6 |
| 24 | Pal et al[67] | CA | Yes | Yes | Yes | No | Yes | Yes | Yes | Yes | No | Yes | 8 |
| 25 | Ricci-Cabello et al[73] | Yes | CA | Yes | Yes | No | Yes | Yes | Yes | Yes | Yes | Yes | 9 |
| 26 | Saffari et al[74] | CA | Yes | Yes | CA | No | Yes | Yes | Yes | Yes | Yes | Yes | 8 |
| 27 | Gucciardi et al[52] | CA | Yes | Yes | No | No | Yes | Yes | Yes | Yes | No | Yes | 7 |
| 28 | Pal et al[68] | Yes | Yes | Yes | Yes | Yes | Yes | Yes | Yes | Yes | No | Yes | 10 |
| 29 | van Vugt et al[75] | CA | Yes | Yes | CA | No | Yes | Yes | Yes | NA | No | Yes | 6 |

Continued

**Table 1** Continued

| | Systematic review/meta-analysis | Item 1 | Item 2 | Item 3 | Item 4 | Item 5 | Item 6 | Item 7 | Item 8 | Item 9 | Item 10 | Item 11 | Total score |
|---|---|---|---|---|---|---|---|---|---|---|---|---|---|
| 30 | Amaeshi[32] | CA | CA | Yes | No | No | Yes | Yes | Yes | NA | No | No | 4 |
| 31 | Nam et al[62] | CA | CA | Yes | Yes | No | Yes | Yes | Yes | Yes | Yes | Yes | 8 |
| 32 | Steinsbekk et al[76] | CA | Yes | Yes | CA | No | Yes | Yes | Yes | Yes | No | Yes | 7 |
| 33 | Burke et al[38] | Yes | Yes | Yes | Yes | Yes | Yes | Yes | Yes | Yes | NA | Yes | 10 |
| 34 | Lun Gan et al[57] | Yes | Yes | Yes | CA | No | Yes | Yes | Yes | Yes | No | Yes | 8 |
| 35 | Ramadas et al[77] | CA | CA | Yes | No | No | Yes | Yes | Yes | NA | No | Yes | 5 |
| 36 | Hawthorne et al[54] | Yes | Yes | Yes | CA | No | Yes | Yes | Yes | Yes | CA | Yes | 8 |
| 37 | Minet et al[61] | Yes | Yes | Yes | No | No | Yes | Yes | Yes | Yes | Yes | Yes | 9 |
| 38 | Alam et al[30] | Yes | Yes | No | CA | No | Yes | Yes | Yes | Yes | Yes | Yes | 8 |
| 39 | Duke et al[46] | Yes | CA | Yes | No | Yes | Yes | Yes | Yes | Yes | No | Yes | 8 |
| 40 | Fan and Sidani[47] | Yes | No | Yes | CA | No | Yes | No | No | Yes | No | Yes | 5 |
| 41 | Hawthorne et al[53] | Yes | Yes | Yes | Yes | Yes | Yes | Yes | Yes | Yes | Yes | Yes | 11 |
| 42 | Khunti et al[58] | CA | Yes | Yes | Yes | No | Yes | No | No | No | No | Yes | 5 |
| 43 | Loveman et al[60] | Yes | CA | Yes | Yes | No | Yes | Yes | Yes | Yes | No | Yes | 8 |
| 44 | Wens et al[78] | CA | Yes | Yes | CA | No | Yes | Yes | Yes | Yes | NA | Yes | 7 |
| 45 | Nield et al[63] | Yes | Yes | Yes | CA | Yes | Yes | Yes | Yes | Yes | No | Yes | 9 |
| 46 | Zabaleta and Forbes[79] | CA | CA | Yes | CA | Yes | Yes | Yes | Yes | NA | No | No | 5 |
| 47 | Deakin et al[43] | Yes | Yes | Yes | Yes | Yes | Yes | Yes | Yes | Yes | Yes | Yes | 11 |
| 48 | Vermeire et al[80] | Yes | Yes | Yes | CA | Yes | Yes | Yes | Yes | Yes | No | Yes | 9 |
| 49 | Gary et al[49] | CA | Yes | No | Yes | No | Yes | Yes | No | Yes | No | Yes | 6 |
| 50 | Norris et al[65] | CA | No | Yes | No | No | Yes | Yes | Yes | CA | No | No | 4 |
| 51 | Norris et al[64] | CA | Yes | Yes | CA | No | Yes | Yes | Yes | NA | No | No | 5 |

Item 1: 'Was an "a priori" design provided?',Source:Shea et al[22]; Item 2: 'Was there duplicate study selection and data extraction?'; Item 3: 'Was a comprehensive literature search performed?'; Item 4: 'Was the status of publication (ie, grey literature) used as an inclusion criterion?'; Item 5: 'Was a list of studies (included and excluded) provided?'; Item 6: 'Were the characteristics of the included studies provided?'; Item 7: 'Was the scientific quality of the included studies assessed and documented?'; Item 8: 'Was the scientific quality of the included studies used appropriately in formulating conclusions?'; Item 9: 'Were the methods used to combine the findings of studies appropriate?'; Item 10: 'Was the likelihood of publication bias assessed?'; Item 11: 'Was the conflict of interest stated?'

CA, cannot answer; NA, not applicable.

Table 2  Characteristics and interventions of included systematic reviews and meta-analysis involved patients with ACS

| First author, year; journal | Primary objectives (to assess effect of interventions on....) | Intervention — Studies details | Intervention — Educational content | Provider | Number of session(s), delivery mode, time, setting | | Outcomes (primary outcomes were in bold) '–': No change '↑': Increase '↓': Decrease | Synthesis methods |
|---|---|---|---|---|---|---|---|---|
| Devi, 2015[44]; *The Cochrane Library* | Lifestyle changes and medicines management | **Number of studies:** 11 completed trials (12 publications); **Types of studies:** RCTs; **Total sample:** 1392 participants | All internet-based interventions | √ BEHA (-) √ CVR (-) √ DIET (-) √ EXERCISE (-) □ MED √ PSY(-) √ SMOKING (-) □ SELF | Dietitians; exercise specialists; nurse practitioners; physiotherapist rehabilitation specialists; nurse, or did not describe. | **Number of session:** weekly or monthly or unclear; **Total contact hours:** unclear. **Duration:** from 6 weeks to 1 year | **Strategies:** internet-based and mobile phone-based intervention, such as email access, private-messaging function on the website, one-to-one chat facility, a synchronised group chat, an online discussion forum, or telephone consultations; or video files; **Format:** one-on-one chat sessions; "ask an expert' group chat sessions; **Theoretical approach:** unclear | Inpatient settings, postdischarge, other | – **Clinical outcomes;** – **Cardiovascular risk factors;** – Lifestyle changes; – Compliance with medication; – Healthcare utilisation and costs; ↓ Adverse intervention effects | Meta-analysis used Review Manager software |
| Barth, 2015[69]; *The Cochrane Library* | Smoking cessation | **Number of studies:**40 RCTs; **Types of studies:** RCTs; **Total sample:** 7928 participants | Psychosocial smoking cessation interventions | □ BEHA □ CVR □ DIET □ EXERCISE □ MED □ PSY √ SMOKING □ SELF | Cardiologist; general practitioner physician or study nurse | **Number of session:** weekly or 2–3 times per week; **Total contact hours:** unclear. **Duration:** from 8 weeks to 1 year | **Strategies:** face-to-face, telephone contact, written educational materials, videotape, booklet or unclear; **Format:** one by one counselling; telephone call; group meetings or unclear; **Theoretical approach:** TTM, SCT | Inpatient settings, postdischarge, other | ↑ **Abstinence by self-report or validated** | Meta- analysis used Review Manager software |
| Kotb, 2014[59]; *PLoS One* | Patients' outcomes | **Number of studies:** 26 studies; **Types of studies:** RCTs; **Total sample:** 4081 participants | Telephone-delivered postdischarge interventions | □ BEHA √ CVR √ DIET □ EXERCISE □ MED □ PSY □ SMOKING □ SELF | Dietitians; exercise specialist; health educators; nurses and pharmacists | **Number of session:** 3–6 sessions/ telephone calls and was greater than six calls in five studies; or unclear; **Total contact hours:** 40 –180 mins or unclear; **Duration:** 1.5– 6 months or unclear | **Strategies:** telephone calls; **Format:** unclear, did not describe the format; **Theoretical approach:** unclear | Unclear, did not describe the setting | ↓ **All-cause hospitalisation;** – **All-cause mortality;** ↓ Depression; – Anxiety; ↑ Smoking cessation; ↓ Systolic blood pressure; – LDL-c | Meta-analysis used Review Manager software |

Continued

**Table 2** Continued

| First author, year; journal | Intervention: Primary objectives (to assess effect of interventions on…) | Studies details | Educational content | Provider | Number of session(s), delivery mode, time, setting | Outcomes (primary outcomes were in bold) '-': No change '↑': Increase '↓': Decrease | Synthesis methods |
|---|---|---|---|---|---|---|---|
| Ghisi, 2014[50], *Patient Education and Counseling* | Knowledge, health behaviour change, medication adherence, psychosocial well-being | **Number of studies:** 42 articles; **Types of studies:** 30 were experimental: 23 RCTs and 7 quasi-experimental; and 11 observational and 1 used a mixed-methods design. **Total sample:** 16 079 participants | Any educational interventions √ BEHA (+) √ CVR (++) √ DIET (+++) √ EXERCISE (++) √ MED (++) √ PSY(++) √ SMOKING (+) □ SELF | Nurses (35.7%), a multidisciplinary team (31%), dietitians (14.3%) and a cardiologist (2.4%) | **Strategies:** did not describe the strategies; **Format:** group (88.1%) education was delivered by lectures (40.5%), group discussions (40.5%) and question and answer periods (7.1%). Individual education (88.1%), including individual counselling (50%), follow-up telephone contacts (31%) and home visits (7.1%); **Theoretical approach:** unclear **Number of session:** 1–24 or unclear. **Total contact hours:** 5–10 min to 3 hours as well as a full day of education **Duration:** 1–24 month; from daily education to every 6 months | – **Knowledge;** – **Behaviour;** – **Psychosocial indicators** | Inpatient settings | Narrative synthesis |
| Brown, 2013[37], *European Journal of Preventive Cardiology* | Mortality, morbidity, HRQoL and healthcare costs | **Number of studies:** 24 papers reporting on 13 RCTs; **Types of studies:** RCTs; **Total sample:** 68 556 participants | Patient education □ BEHA √ CVR □ DIET □ EXERCISE □ MED □ PSY □ SMOKING □ SELF | Nurses or other healthcare professionals. | **Strategies:** face-to-face education sessions, telephone contact and interactive use of the internet; **Format:** group-based sessions, individualised education and four used a mixture of both sessions; **Theoretical approach:** unclear **Number of session and duration:** from a total of 2 visits to a 4-week residential stay reinforced with 11 months of nurse led follow-up **Total contact hours:** unclear | – **Mortality,** – **Non-fatal MI,** – **Revascularisations,** – **Hospitalisations,** – **HRQoL,** – Withdrawals/ dropouts; – Healthcare utilisation and costs | Inpatient settings, other | Meta- analysis used Review Manager software |
| Dickens, 2013[45], *Psychosomatic Medicine* | Depression and depressive symptoms | **Number of studies:** 62 independent studies **Types of studies:** RCTs; **Total sample:** 17 397 | Psychological interventions √ BEHA (-) □ CVR □ DIET □ EXERCISE □ MED √ PSY (-) □ SMOKING √ SELF (-) | A single health professional or by a unidisciplinary team | **Strategies:** face-to-face sessions, telephone contact or unclear; **Format:** group or unclear; **Theoretical approach:** unclear **Number of session:** 14.4 (range, 1–156); **Total contact hours:** varying from 10 to 240 min **Duration:** unclear | ↓ **Depression;** – Adverse cardiac outcomes; – Ongoing cardiac symptoms | Unclear, did not describe | Univariate analyses using comprehensive meta-analysis, multivariate meta-regression using SPSS V.15.0 |

Continued

**Table 2** Continued

| First author, year; journal | Primary objectives (to assess effect of interventions on....) | Intervention — Studies details | Intervention — Educational content | Educational content | Provider | Number of session(s), delivery mode, time, setting | setting | Outcomes (primary outcomes were in bold) '–': No change '↑': Increase '↓': Decrease | Synthesis methods |
|---|---|---|---|---|---|---|---|---|---|
| Aldcroft, 2011[20]; Journal of Cardiopulmonary Rehabilitation & Prevention | Health behaviour change | **Number of studies:** seven trials; **Types of studies:** six randomised controlled trials and a quasi-experimental trial; **Total sample:** 536 participants | All psychoeducational or behavioural intervention | □ BEHA; √ CVR (-); □ DIET; □ EXERCISE; □ MED; √ PSY (-); □ SMOKING; □ SELF | Appropriately trained healthcare workers | **Strategies:** did not describe the strategies; **Format:** group setting, combination of group and one-on-one education and one-on-one format only; **Theoretical approach:** TTM, interactionist role theory, Bandura's self-efficacy theory, Gordon's relapse prevention model and a cognitive behavioural approach. **Number of session:** unclear; **Total contact hours:** unclear; **Duration:** 2–12 months | Unclear, did not describe | ↓ **Smoking rates; medication use;** **– Supplemental oxygen use;** ↑ **Physical activity;** ↑ Nutritional habits | Meta-analysis and narrative presentation |
| Brown, 2011[70]; The Cochrane Library | Mortality, morbidity, HRQoL and healthcare costs | **Number of studies:** 24 papers reporting on 13 studies. **Types of studies:** RCTs; **Total sample:** 68 556 participants | Patient education | √ BEHA (-); √ CVR (-); □ DIET; √ EXERCISE (-); √ MED; □ PSY; □ SMOKING; □ SELF | Nurse or did not describe | **Strategies:** face-to-face sessions, telephone contact and interactive use of the internet; **Format:** four studies involved group sessions, five involved individualised education and three used both session types, with one study comparing the two approaches; **Theoretical approach:** did not describe. **Number of session and duration:** two visits to 4 weeks residential 11 months of nurse led follow-up. **Total contact hours:** unclear | Postdischarge, other | – **Total mortality;** – **Cardiovascular mortality;** – **Non-cardiovascular mortality;** – **Total cardiovascular (CV) events;** – **Fatal and/or non-fatal MI;** – Other fatal and/or non-fatal CV events | Meta-analysis used Review Manager software |
| Goulding, 2010[51]; Journal of Advanced Nursing | Change maladaptive illness | **Number of studies:** 13 studies; **Types of studies:** RCTs; **Total sample:** unclear | Interventions to change maladaptive illness beliefs | √ BEHA (-); □ CVR; □ DIET; □ EXERCISE; □ MED; √ PSY (-); □ SMOKING; □ SELF | Cardiologist, nurse, psychologist or did not describe. | **Strategies:** face-to-face sessions, telephone contact and written self-administered; **Format:** unclear; **Theoretical approach:** Common Sense Model, Leventhal's framework. **Number of session:** unclear; **Total contact hours:** unclear; **Duration:** 4 days to 2 weeks or unclear | Inpatient settings, postdischarge, other | – **Beliefs (or other illness cognition);** – QoL; – Behaviour; – Anxiety or depression; – Psychological well-being; – Modifiable risk factors; protective factors | A descriptive data synthesis |
| Huttunen-Lenz, 2010[56]; British Journal of Health Psychology | Smoking cessation | **Number of studies:** a total of 14 studies were included **Types of studies:** RCTs; **Total sample:** 1792 participants | Psychoeducational cardiac rehabilitation intervention | □ BEHA; □ CVR; □ DIET; □ EXERCISE; □ MED; □ PSY; √ SMOKING (-); □ SELF | Cardiologist, nurse psychologist or did not describe | **Strategies:** face-to-face counselling, self-help materials; home visit, booklet, video and telephone contact; **Format:** individual or unclear; **Theoretical approach:** social learning theory; ASE model; TTM; behavioural multicomponent approach. **Number of session:** 4–20 or unclear. **Total contact hours:** 10–720 mins or unclear; **Duration:** 4–29 weeks or unclear | Inpatient settings, postdischarge, other | ↑ **Prevalent smoking cessation,** ↑ **Continuous smoking cessation,** – Mortality | Subgroup meta-analysis was used software |

**Table 2** Continued

| First author, year/journal | Primary objectives (to assess effect of interventions on....) | Intervention — Studies details | Intervention — Educational content | Provider | Number of session(s), delivery mode, time, setting | Outcomes (primary outcomes were in bold) '–': No change '↑': Increase '↓': Decrease | Synthesis methods |
|---|---|---|---|---|---|---|---|
| Auer, 2008[34]; *Circulation* | Multiple cardiovascular risk factors and all-cause mortality | **Number of studies:** 27 articles reporting 26 studies **Types of studies:** 16 clinical controlled trials and 10 before-after studies **Total sample:** 2467 patients in CCTs and 38, 581 patients in before-after studies | In-hospital multidimensional interventions of secondary prevention<br>□ BEHA<br>□ CVR<br>√ DIET (-)<br>√ EXERCISE (-)<br>√ MED<br>√ PSY (-)<br>√ SMOKING (-)<br>□ SELF | Cardiac nurses; physician, or did not describe | **Number of session:** 1–5 or unclear; **Total contact hours:** 30–240 mins or unclear; **Duration:** 4 weeks–12 months. **Strategies:** Written material; audiotapes; presentations; face-to-face; **Format:** group or unclear; **Theoretical approach:** unclear. Inpatient settings | ↓ **All-cause mortality;** ↓ **Readmission rates;** – **Reinfarction rates** | Stata V.9.1 |
| Barth, 2008[36]; *The Cochrane Library* | Smoking cessation | **Number of studies:** 40 trials; **Types of studies:** RCTs; **Total sample:** 7682 patients | Psychosocial intervention<br>√ BEHA (+++)<br>√ CVR (++)<br>□ DIET<br>□ EXERCISE<br>□ MED<br>√ PSY (+)<br>√ SMOKING (+++)<br>√ SELF(+++) | Cardiologist, nurse, physician or study nurse | **Number of session:** 1–5 or unclear; **Total contact hours:** 15 mins–9 hours **Duration:** within 4 weeks or did not report on the duration. **Strategies:** face-to-face; information booklets, audiotapes or videotapes **Format:** group sessions or individual counselling; **Theoretical approach:** TTM. Inpatient settings | ↑ **Abstinence by self-report or validated** | Meta-analysis used Review Manager software |
| Fernandez, 2007[48]; *International Journal of Evidence-Based Healthcare* | Risk factor modification | **Number of studies:** 17 trials; **Types of studies:** randomised, quasi-RCTs and clustered trials; **Total sample:** 4725 participants | Brief structured intervention<br>√ BEHA (-)<br>√ CVR (-)<br>□ DIET<br>□ EXERCISE<br>□ MED<br>□ PSY<br>□ SMOKING<br>√ SELF (-) | Case manager; dieticians; health educator; nurses; psychologist; and research assistants | **Number of session:** supportive counselling ranged from 1 to 7 calls for the duration of the study; **Total contact hours:** varied from 10 to 30 mins; **Duration:** unclear. **Strategies:** written, visual, audio, telephone contact; **Format:** did not describe; **Theoretical approach:** theoretical behaviour change principles. Unclear, did not describe | ↓ **Smoking;** – **Cholesterol level;** – **Physical activity;** ↑ **Dietary habits;** – **Blood sugar levels;** – **BP levels;** ↓ **BMI;** – Incidence of admission | Cochrane statistical package Review Manager |
| Barth, 2006[35]; *Annals of Behavioural Medicine* | Smoking cessation | **Number of studies:** 19 trials; **Types of studies:** RCTs; **Total sample:** 2548 patients | Psychosocial interventions<br>√ BEHA (+++)<br>√ CVR (++)<br>□ DIET<br>□ EXERCISE<br>□ MED<br>□ PSY<br>□ SMOKING<br>√ SELF (+++) | Unclear, did not describe | **Number of session:** unclear; **Total contact hours:** unclear; **Duration:** unclear. **Strategies:** face-to-face, telephone contact or unclear; **Format:** unclear; **Theoretical approach:** unclear. Unclear, did not describe | ↑ **Abstinence;** ↓ **Smoking status** | Data analyses were carried out in Review Manager V.4.2 |

Continued

**Table 2** Continued

| First author, year; journal | Primary objectives (to assess effect of interventions on…..) | Intervention | | | | | Outcomes (primary outcomes were in bold) '–': No change '↑': Increase '↓': Decrease | Synthesis methods |
|---|---|---|---|---|---|---|---|---|
| | | Studies details | Educational content | Provider | Number of session(s), delivery mode, time, setting | | | |
| Clark, 2005[41]; Annals of Internal Medicine | Mortality, MI | Number of studies: 63 randomised trials; Types of studies: RCTs; Total sample: 21 295 patients | Secondary prevention programmes | □ BEHA □ CVR √ DIET (–) √ EXERCISE (–) □ MED √ PSY (–) □ SMOKING □ SELF | Nurse, multidisciplinary team or did not describe | Number of session: 1–12 or unclear Total contact hours: did not describe Duration: 0.75–48 months | Strategies: face-to-face, telephone contact and home visit; Format: group and individual or unclear; Theoretical approach: unclear | Inpatient settings, postdischarge, other | ↓ **Mortality**, ↓ **MI**, – Hospitalisation rates | Performed analyses by using Review Manager V.4.2 and Qualitative Data Synthesis |

Smoking, smoking cessation; CVR, cardiovascular risk factors; PSY, psychosocial issues (depression, anxiety); DIET, diet; EXERCISE, exercise; MED, medication; BEHA, behavioural change (including lifestyle modification); SELF, self-management (including problems solving); DR, diabetes risks; CHD, coronary heart disease; CAD, coronary artery disease; CHW, community health worker; HbA1c, glycated haemoglobin; BP, blood pressure; LDL, low-density lipoprotein cholesterol; SMS, short message service; BCTs, behavioural change techniques; LEA, lower extremity amputation; PRIDE, Problem Identification, Researching one's routine, Identifying a management goal, Developing a plan to reach it, Expressing one's reactions and Establishing rewards for making progress; ASE, attitude social influence-efficacy; CVRF, cardiovascular risk factors; PA, physical activity; EDU, patient education; GP, general practice; RCTs, randomised controlled trials; CCTS, controlled clinical trials; HRQoL, health-related quality of life; QoL, quality of life; MI, myocardial infarction; CAD, coronary artery disease; CABG, coronary artery bypass graft surgery; BMI, body mass index; SBP, systolic blood pressure; DBP, diastolic blood pressure; HDL-c, high-density lipoprotein cholesterol; TTM, transtheoretical model; SCT, social cognitive theory; HBM, health belief model; SAT, social action theory.

In the educational content: '+': minor focus; '++': moderate focus; '+++': major focus; '–' =unclear what the intensity of the education was for any topic.

In the outcomes: arrow up ('↑') for improvement, arrow down ('↓') for reduction; a dash ('–') for no change or inconclusive evidence. Primary outcomes were in bold.

25.6 (range, 7–132) studies was included per systematic review or meta-analyses. Of the total, 818 unique (non-repeated) studies were included in all of the reviews or meta-analyses, 286 included patients with ACS and 532 included patients with T2DM (see online supplementary appendix 2 and 3). The included reviews assessed the risk of bias using the Cochrane risk of bias tool (22 publications), JADA quality score (7 publications), Joanna Briggs quality assessment tool (2 publications), PEDro scale (1 publication), RCT Critical Appraisal Skills Programme (1 publication) and the SIGN-50 checklist (1 publication).

## Methodological quality of included systematic reviews and meta-analyses

The methodological quality of the included publications is presented in table 1. Thirty (58.8%) publications were classified as high quality (scores 8–11) and 21 (41.2%) publications were classified as medium quality (scores 4–7). Twenty-five (49%) reviews specifically provided an a priori design, while the use of such a design was unclear for 26 (51%) publications. The inclusion of other forms of literature (such as grey literature) was described in 18 (35%) reviews. Only 14 out of 51 (27%) reviews included a table of included and excluded studies. Only two (4%) reviews did not provide a characteristics table of the included papers. The scientific quality of the included papers was evaluated and documented in 47 (92%) reviews. The scientific quality of the included studies was used appropriately to formulate conclusions in 47 (92%) reviews. The methods to combine the results of the included studies were appropriate in 43 (86%) reviews. Publication bias was assessed in only 19 (37%) reviews. Finally, conflicts of interest were reported in 47 (92%) reviews.

## Characteristics of health educational interventions

The description of the health educational interventions followed the Workgroup for Intervention Development and Evaluation Research reporting guidelines for behaviour change interventions.[81] The characteristics of the recipients, setting, delivery methods, intensity, duration and educational content of health educational interventions for patients with ACS or T2DM are summarised in tables 2 and 3. The delivery strategies for health education included face-to-face, internet-based, phone-based, videotape, written educational materials or mixed. The format included one-on-one (individualised), group or both. Face-to-face sessions were the most common delivery formats, and many education sessions were also delivered by telephone/web contact or individualised counselling. The number of sessions, total contact hours and durations varied, and there was limited information about the intensity of health education for patients provided. The frequency of educational sessions was weekly or monthly, and an average of 3.7 topics was covered per education session. Nurses and multidisciplinary teams were the most frequent educators, and most education programmes were delivered postdischarge.

**Table 3** Characteristics and interventions of included systematic reviews and meta-analysis involved patients with T2DM

| First author; year; journal | Intervention — Primary objectives (to assess effect of interventions on….) | Studies details | Educational content | Intervention | Provider | Number of session(s), delivery mode, time, setting | Setting | Outcomes (primary outcomes were in bold.) –: No change ↑: Increase ↓: Decrease | Synthesis methods |
|---|---|---|---|---|---|---|---|---|---|
| Choi, 2016[40]; Diabetes Research and Clinical Practice | Glycaemic effect | **Number of studies:** 53 studies (5 in English, 48 in Chinese); **Types of studies:** RCTs; **Total sample:** unclear | Diabetes education intervention | □ BEHA; √ DIET (-); □ DR; □ EXERCISE; □ GC; □ MED; □ PSY; □ SMOKING; √ SELF (-) | Unclear, did not describe | **Number of session:** unclear; **Total contact hours:** unclear; **Duration:** 30–150min or unclear; **Strategies:** face-to-face, written materials; telephone contact and home visit; **Format:** unclear; **Theoretical approach:** unclear | Inpatient settings, post discharge, other | ↓ **HbA1c**, | STATA V.12 and Review Manager V.5.3 |
| Creamer, 2016[42]; Diabetic Medicine | Successful outcomes and to suggest directions for future research | **Number of studies: 33;** **Types of studies:** RCTs; **Total sample:** 7453 participants | Culturally appropriate health education | √ BEHA (-); √ DIET (-); √ DR (-); √ EXERCISE; □ GC; □ MED; □ PSY; □ SMOKING; √ SELF (-) | CHWs, clinical pharmacists dieticians, nurses, podiatrists, physiotherapists and psychologists | **Number of session:** 1–10 or unclear; **Total contact hours:** unclear; **Duration:** from a single session to 24months; **Strategies:** face-to-face; phone contact; **Format:** group sessions (13 studies), individual sessions (10 studies) or a combination of both; **Theoretical approach:** unclear | Inpatient settings, postdischarge, other | ↓ **HbA1c,** – **HRQoL,** – **Adverse events,** – BP, – BMI, – Lipid levels, – Diabetes complications, – Economic analyses, mortality and diabetes knowledge, – Empowerment, – Self-efficacy and satisfaction | Meta-analysis using the Review Manager statistical programme |
| Huang, 2016[55]; European Journal of Internal Medicine | Clinical markers of cardiovascular disease | **Number of studies:** 17 studies; **Types of studies:** RCTs; **Total sample:** unclear | Lifestyle interventions | □ BEHA; √ DIET (-); √ CVR (-); √ EXERCISE (-); □ GC; □ MED; □ PSY; □ SMOKING; √ SELF (-) | Nurse, pharmacist or unclear | **Number of session:** unclear; **Total contact hours:** unclear; **Duration:** 6months–8years; **Strategies:** unclear; **Format:** individual; group and mixed; **Theoretical approach:** unclear | Unclear, did not describe | **Cardiovascular risk factors** such as, – BMI, – HbA1c, – BP, ↓ Level of cholesterol | Review Manager V.5.1 |
| Chen, 2015[39]; Metabolism- Clinical and Experimental | Clinical markers | **Number of studies:** 16 studies; **Types of studies:** RCTs; **Total sample:** per study ranged from 23 to 2575 | Lifestyle intervention | √ BEHA (-); □ DIET; √ CVR (-); □ EXERCISE; □ GC; √ MED (-); □ PSY; □ SMOKING; √ SELF (-) | Unclear, did not describe | **Number of session:** monthly; **Total contact hours:** unclear; **Duration:** <6months–8years; **Strategies:** unclear; **Format:** individual; group and mixed; **Theoretical approach:** unclear | Unclear, did not describe | **Cardiovascular risk factors** including – BMI, ↓ HbA1c, ↓ SBP, DBP, ↓ HDL–c and LDL–c | All analyses were performed using Comprehensive Meta-Analysis statistical software |
| Terranova, 2015[72]; Diabetes, Obesity and Metabolism | Weight loss | **Number of studies:** 10 individual studies (from 13 papers); **Types of studies:** RCTs; **Total sample:** ranging from 27 to 5145 participants | Lifestyle-based-only intervention | √ BEHA (-); √ DIET (-); □ DR; √ EXERCISE (-); □ GC; □ MED; □ PSY; □ SMOKING; √ SELF (-) | Dietician; diabetes educator; general physician; multidisciplinary team or nutritionist; nurse | **Number of session:** 1–42; **Total contact hours:** unclear; **Duration:** ranged from 16 weeks to 9years; **Strategies and format:** face-to-face individual or group-based sessions, or a combination of those. One study delivered the intervention via the telephone; **Theoretical approach:** unclear | Unclear, did not describe | ↓ **Weight change;** – **HbA1c** | Meta-analyses—Review Manager and meta-regression analysis—Stata version. |
| Pillay, 2015[71]; Annals of Internal Medicine | HbA1c level | **Number of studies:** 132; **Types of studies:** RCTs; **Total sample:** unclear | Behavioural programme | √ BEHA (-); √ DIET (-); □ DR; √ GC (-); √ MED (-); □ PSY; □ SMOKING; √ SELF (-) | Trained individuals | **Number of session:** unclear; **Total contact hours:** range, 7–40.5hours; **Duration:** 4 or more weeks | Inpatient settings, post discharge, other | – **HbA1c;** ↓ **BMI** | The analysis was conducted by using a Bayesian network model |

**Table 3** Continued

| First author, year; journal | Primary objectives (to assess effect of interventions on...) | Intervention: Studies details | Intervention: Educational content | Intervention: Behaviours | Provider | Number of session(s), delivery mode, time, setting | Setting | Outcomes (primary outcomes were in bold.) –: No change ↑: Increase ↓: Decrease | Synthesis methods |
|---|---|---|---|---|---|---|---|---|---|
| Pal, 2014[87]; Diabetes Care | Health status, cardiovascular risk factors and QoL | Number of studies: 20 papers describing 16 studies; Types of studies: RCTs; Total sample: 3578 participants | Computer-based self-management interventions | □BEHA; □DIET; □DR; □EXERCISE; □GC; □MED; □PSY; □SMOKING; √SELF | Unclear, did not describe | Number of session: 1–8; Total contact hours: 10 min – 6 hours; Duration: 8 weeks–12 months | Strategies: online/web-based; Phone contact; Format: individual; group and mixed; Theoretical approach: TTM, social ecological theory, SCT and self-determination theory — Unclear, did not describe | –HRQoL; ↓HbA1c; –Death; ↓Cognitions, behaviours, –Social support, ↓Cardiovascular risk factors, –Complications, –Emotional outcomes, –Hypoglycaemia, –Adverse effects, –CE and economic data | Meta-analysis using Review Manager software or narrative presentation |
| Ricci-Cabello, 2014[73]; BMC Endocrine Disorders | Knowledge, behaviours and clinical outcomes | Number of studies: 37 studies; Types of studies: almost two-thirds of the studies were RCTs, 27% studies were quasi-experimental design. Total sample: unclear | DSM educational programme | □BEHA; √DIET(+++); □DR; √EXERCISE (+++); √GC(+++); √MED(++); √PSY(++); □SMOKING; □SELF | Dietitian; nurse; psychologist; physician; research team or staff | Number of session: 13.1; Total contact hours: 0.25–180 hours; Duration: 0.25–48 months | Strategies: face-to-face; telecommunication; both; Format: one on one; group and mixed; Theoretical approach: unclear — Postdischarge, other | –Diabetes knowledge; –Self-management; –Behaviours; –Clinical outcomes; ↓Glycated haemoglobin; –Cost-effectiveness analysis | Meta-analyses and bivariate meta-regression were conducted with Stata V.12.0 |
| Saffari, 2014[74]; Primary Care Diabetes | Glycaemic control. | Number of studies: 10; Types of studies: RCTs; Total sample: 960 patients | An educational intervention using SMS | √BEHA (-); □DIET; □DR; □EXERCISE; √GC (-); √MED (-); □PSY; □SMOKING; □SELF | Unclear, did not describe | Number of session: weekly; or two messages daily or unclear; Total contact hours: unclear; Duration: 3 months–1 year | Strategies: SMS: sending and receiving data. Receive data through text-messaging by patients only. Used a website along with SMS; Format: Unclear; Theoretical approach: Unclear. — Inpatient settings, postdischarge, other | ↑Glycaemic control | Comprehensive Meta-analysis Software V.2.0 |
| Odnoletkova, 2014[66]; Journal of Diabetes & Metabolism | Cost-effectiveness (CE) | Number of studies: 17 studies; Types of studies: RCTs; Total sample: unclear | Therapeutic education | √BEHA (-); □DIET; □DR; □EXERCISE; □GC; □MED; □PSY; □SMOKING; √SELF (-) | General physician; nutritionists or unclear | Number of session: ~16; Total contact hours: unclear; Duration: unclear | Strategies: face-to-face or unclear; Format: individual and group lessons; Theoretical approach: unclear — Inhospital or unclear | –CE | Incremental cost-effectiveness ratio |
| Attridge, 2014[33]; The Cochrane Library | HbA1c level, knowledge and clinical outcomes | Number of studies: 33 trials; Types of studies: RCTs and quasi-RCTs; Total sample: 7453 participants | 'Culturally appropriate' health education | √BEHA (-); √DIET (-); □DR; √EXERCISE (-); √GC (-); □MED; □PSY; √SMOKING (-); □SELF | CHWs; dieticians; exercise physiologists; lay workers; nurses; podiatrists and psychologists | Number of session: one session to 24 months; Total contact hours: unclear; Duration: the median duration of interventions was 6 months | Strategies: Format: group intervention method, one-to-one sessions and a mixture of the two methods. Or a purely interactive patient-centred method; Theoretical approach: empowerment theories; behaviour change theories, TTM of behaviour change and SCT — Inpatient settings, postdischarge, other | ↓HbA1c; –HRQoL; –Adverse events; –Mortality; –Complications; –Satisfaction; ↑Empowerment; ↑Self-efficacy; –Attitude; knowledge; –BP; –BMI; –Lipid levels; –Health economics | Meta-analyses used Review Manager software |
| Vugt, 2013[75]; Journal of Medical Internet Research | Health outcomes | Number of studies: 13 studies; Types of studies: RCTs; Total sample: 3813 patients | BCTs are being used in online self-management interventions | √BEHA (-); □DIET; □DR; □EXERCISE; □GC; □MED; □PSY; □SMOKING; √SELF (-) | Healthcare professional | Number of session: 6 weekly sessions or unclear; Total contact hours: unclear; Duration: unclear | Strategies: online/web-based; Format: unclear; Theoretical approach: self-efficacy theory, social support theory, TTM, SCT, social-cognitive model and cognitive behavioural therapy — Postdischarge | –Health behaviour change; –Psychological well-being; –Clinical parameters | Unclear |

Continued

**Table 3** Continued

| First author, year; journal | Primary objectives (to assess effect of interventions on....) | Studies details | Intervention: Educational content | Intervention | Provider | Number of session(s), delivery mode, time, setting | Outcomes (primary outcomes were in bold.) '–': No change '↑': Increase '↓': Decrease | Synthesis methods |
|---|---|---|---|---|---|---|---|---|
| Gucciardi, 2013[52]; Patient Education and Counselling | HbA1c level, physical activity and diet outcomes | **Number of studies:** 13 studies; **Types of studies:** RCTs and comparative studies; **Total sample:** unclear | DSME interventions. | □ BEHA; √ DIET (+++); □ DR; √ EXERCISE (+++); □ GC; √ MED (+); √ PSY (+); □ SMOKING; √ SELF (++) | Dietitians (n=7/13); Multidisciplinary team (n=7/13); Nurse (n=5/13); Community peer worker (n=3/13) | **Strategies:** face-to-face (n=13/13); written literature: (eg, handbook) (n=4/13); telephone (n=7); high intensity: ≥10 education sessions (n=6); **Format:** one-on-one: (n=11/13); group (n=9/13); **Theoretical approach:** SAT; empowerment Behaviour change model; modification theories; pharmaceutical care model; Behaviour change theory; PATHWAYS programme; symptom-focused management model; motivational interviewing. **Number of session:** low intensity: <10 education sessions (n=7); high intensity: ≥10 education sessions (n=6); **Total contact hours:** unclear; **Duration:** <6months (n=7/13); ≥6months (n=6/13). Inpatient settings, postdischarge | – HbA1c levels, – Anthropometrics, – Physical activity, – Diet outcomes | A recently described method |
| Pal, 2013[68], The Cochrane Library | Health status and HRQoL | **Number of studies:** 16 studies; **Types of studies:** RCTs; **Total sample:** 3578 participants | Computer-based diabetes self-management intervention | □ BEHA; √ DIET (-); □ DR; √ EXERCISE (-); √ GC (-); √ MED (-); √ PSY (-); □ SMOKING; □ SELF | Nurse or other healthcare professionals | **Strategies:** online/web-based; phone contact; **Format:** unclear; **Theoretical approach:** unclear. **Number of session:** unclear; **Total contact hours:** unclear; **Duration:** 1 session–18months. Inpatient settings, postdischarge, other | – HRQoL; – **Death from any cause;** ↓**HbA1c;** – Cognitions; – Behaviours; – Social support; – Biological markers; – Complications | Formal meta-analyses and narrative synthesis |
| Nam, 2012[62], Journal of Cardiovascular Nursing | Glycaemic control | **Number of studies:** 12 RCTs; **Types of studies:** RCTs; **Total sample:** 1495 participants | Diabetes educational interventions (no drug intervention) | □ BEHA; √ DIET (-); □ DR; √ EXERCISE (-); √ GC (-); √ MED (-); √ PSY (-); □ SMOKING; √ SELF (-) | Nurses (36%), dieticians (36%), diabetes educators (5%), other professionals (9%) and non-professional staff (14%) | **Strategies:** teaching or counselling; home-based support and visual aids **Format:** group education or a combination of group education and individual counselling; or only individual counselling; **Theoretical approach:** unclear. **Number of session:** 1 month or less; 1–3months; or 12months; most studies did not describe, or from 1 session to more than 30hours; **Duration:** from 1 session to 12months, frequency: 1 session to 25 weekly or biweekly education. Inpatient settings, postdischarge, other | ↓**HbA1c level** | Meta-analysis |
| Steinsbekk, 2012[76]; BMC Health Services Research | Clinical, lifestyle and psychosocial outcomes | **Number of studies:** 21 studies (26 publications) **Types of studies:** RCTs; **Total sample:** 2833 participants | Group-based education | Did not describe the content of the intervention | Community workers; dietician; lay health advisors nurse and nutritionist | **Strategies:** face-to-face; **Format:** 5 to 8 participants group to 40 patients group **Theoretical approach:** empowerment model and the discovery learning theory, the SCT and the social ecological theory, the self-efficacy and self-management theories and operant reinforcement theory. **Number of session and total contact hours:** 30hours over 2.5months, 52 hours over 1 year and 36 or 96hours over 6 months **Duration:** 6 months to 2 years. Inpatient settings, postdischarge, other | ↓**HbA1c,** ↑**Lifestyle outcomes,** ↑**Diabetes knowledge,** ↑Self-management skills, ↓Psychosocial outcomes, ↓Mortality rate, ↓BMI, ↓Blood pressure; ↓Lipid profile | Meta-analysis using Review Manager V.5 |
| Amaeshi, 2012[32], Podiatry Now | Increasing good foot health practices that will ultimately reduce LEA | **Number of studies:** eight studies; **Types of studies:** RCT or clinical controlled trial (CCT); **Total sample:** unclear | Foot health education | Food care | Podiatrist, psychologist or unclear | **Strategies:** face-to-face; **Format:** in three of the studies, educational interventions were delivered to the participants in groups, while the other five provided individualised (one-to-one) foot care education to the participants; **Theoretical approach:** unclear. **Number of session:** unclear; **Total contact hours:** between 15min and 14hours; **Duration:** 3–30months. Unclear, did not describe | ↓LEA; ↑Self-care | Narrative synthesis |

Continued

**Table 3** Continued

| First author, year; journal | Primary objectives (to assess effect of interventions on...) | Intervention — Studies details | Intervention — Educational content | Intervention content | Number of session(s), delivery mode, time, setting | Provider | Outcomes (primary outcomes were in bold) -: No change; ↑: Increase; ↓: Decrease | Synthesis methods |
|---|---|---|---|---|---|---|---|---|
| Lun Gan, 2011[57]; *JBI Library of Systematic Reviews* | Oral hypoglycaemic adherence | **Number of studies:** seven studies; **Types of studies:** RCTs; **Total sample:** unclear | Educational interventions | √ BEHA (-); √ DIET (-); □ DR; √ EXERCISE (-); □ GC; √ MED (-); √ PSY (-); □ SMOKING; √ SELF (-) | **Number of session:** 1–12 or unclear; **Total contact hours:** 2.5 hours or unclear; **Duration:** 4–12 months; **Strategies:** face-to-face; **Format:** group and individual; **Theoretical approach:** unclear. Inpatient settings, postdischarge, other | Nurses; pharmacists; other skilled healthcare professionals | ↓ HbA1c, ↓ **Medication adherence**; ↓Blood glucose; - Tablet count; - Medication containers; - Diabetes complications; - Health service utilisation | Narrative summary form |
| Burke, 2011[38]; *JBI Database of Systematic Reviews and Implementation Reports* | HbA1c level, BP | **Number of studies:** 11 RCTs and 4 quasi-experimental trials; **Types of studies:** RCTs and quasi-experimental trials; **Total sample:** 2240 patients | Group medical visits | √ BEHA (-); √ DIET (-); □ DR; □ EXERCISE; √ GC (-); √ MED (+); □ PSY; □ SMOKING; √ SELF (-) | **Number of session:** 1–4 or unclear; **Total contact hours:** 2–4 hours or unclear; **Duration:** 1 session to 2 years; **Strategies:** face-to-face; **Format:** group and individual; **Theoretical approach:** unclear. Inpatient settings, postdischarge, other | Endocrinologists; DM nurse; family physician; nutritionist and rehab therapist | ↓HbA1c; -**Systolic and diastolic BP**; -LDL measurements | Meta-analysis |
| Ramadas, 2011[77]; *International Journal of Medical Informatics* | HbA1c level | **Number of studies:** 13 different studies; **Types of studies:** RCTs and quasi-experimental studies; **Total sample:** unclear | Web-based behavioural interventions | √ BEHA (-); √ DIET (-); □ DR; □ EXERCISE; √ GC (-); √ MED (+); □ PSY; □ SMOKING; √ SELF (-) | **Number of session:** unclear; **Total contact hours:** unclear; **Duration:** ranged between 12 and 52 weeks, with an average of 27.2±18.3 weeks; **Strategies:** email and SMS technologies that were commonly used together with the websites to reinforce the intervention, and website, print material; **Format:** unclear; **Theoretical approach:** Wagner's Chronic Care Model; self-efficacy theory/social support theory; TTM; HBM; SCT. Inpatient settings, postdischarge, other | Dietician; endocrinologist; physicians; researchers or research staff members and study nurse | - **Self-monitoring blood sugar**, - Weight loss, - Dietary behaviour, - Physical activity | Not statistically combined and re-analysed |
| Minet, 2010[61]; *Patient Education and Counseling* | Glycaemic control | **Number of studies:** 47 studies; **Types of studies:** RCTs; **Total sample:** unclear | Self-care management interventions | √ BEHA (-); □ DIET; □ DR; □ EXERCISE; □ GC; □ MED; □ PSY; □ SMOKING; √ SELF (-) | **Number of session:** 3–26; **Total contact hours:** unclear; **Duration:** 4 weeks to 4 years; **Strategies:** face-to-face; phone calls; home visit; **Format:** group and individual; **Theoretical approach:** unclear. Inpatient settings, postdischarge, other | Case nurse manager; group facilitator; nurse educator; multidisciplinary team; physiologist; physician; peer counsellor; researcher and pharmacist | ↓ **HbA1c** | Meta-analyses and meta-regression used Stata's meta command |
| Hawthorne, 2010[54]; *Diabetic Medicine* | Effects of culturally appropriate health education | **Number of studies:** 10 trials; **Types of studies:** RCTs; **Total sample:** 1603 patients | Culturally appropriate health education | □ BEHA; √ DIET (-); □ DR; √ EXERCISE (-); □ GC; □ MED; □ PSY; √ SMOKING; √ SELF (-) | **Number of session:** unclear; **Total contact hours:** unclear; **Duration:** 1 session to 12 months; **Strategies:** face-to-face; visual aids, leaflets and teaching materials; **Format:** group approach, one-to-one interviews and a mixed approach; **Theoretical approach:** SAT, Empowerment Behaviour Change Model, SCT, Management model and the Theory of Planned Behaviour. Inpatient settings, postdischarge, other | Exercise physiologists; dieticians; diabetes nurses; link workers and podiatrists | -QoL; ↓HbA1c; -**BP**; ↑Knowledge; -BMI; ↓ Lipid levels, - Diabetic complications, - Mortality rates, hospital admissions, hypoglycaemia | Meta-analysis using the Review Manager and narrative review |
| Fan, 2009[47]; *Canadian Journal of Diabetes* | Knowledge, self-management behaviours and metabolic control | **Number of studies:** 50 studies; **Types of studies:** RCTs; **Total sample:** unclear | DSME intervention | √ BEHA (-); □ DIET; □ DR; □ EXERCISE; □ GC; □ MED; √ PSY (-); □ SMOKING; √ SELF (-) | **Number of session:** 10 (range 1–28); **Total contact hours:** 17 contact hours (range 1–52); ≤10 (46%); 11–20 (21%); >20 (33%); **Duration:** 22 weeks (range 1–48); ≤8 weeks (26%); 9–24 weeks (37%); >24 weeks (37%); **Strategies:** Online/web-based (4%); video (2%); face-to-face (60%); phone contact (4%); Mixed (30%); **Format:** one-on-one (32%); group (40%); mixed (28%); **Theoretical approach:** unclear. Inpatient settings, postdischarge, other | Unclear, did not describe | ↑ **Diabetes knowledge**, ↑ **Self-management behaviours**; ↓HbA1c | Comprehensive meta-analysis (V.2.0) |

Continued

**Table 3** Continued

| First author, year; journal | Primary objectives (to assess effect of interventions on…) | Intervention: Studies details | Intervention: Educational content | | Provider | Number of session(s), delivery mode, time, setting | | Outcomes (primary outcomes were in bold.) –: No change ↑: Increase ↓: Decrease | Synthesis methods |
|---|---|---|---|---|---|---|---|---|---|
| Duke, 2009[46]; *The Cochrane Library* | Metabolic control, diabetes knowledge and psychosocial outcomes | **Number of studies:** nine studies; **Types of studies:** RCTs; **Total sample:** 1359 participants | Individual patient education | √ BEHA (-); □ DIET; □ DR; √ EXERCISE (-); √ GC (-); □ MED; □ PSY (-); □ SMOKING; □ SELF | Diabetes educators and dieticians | **Number of session:** 1–6; **Total contact hours:** 20min–7hours; **Duration:** 4 weeks–1 year | **Strategies:** face to face; telephone; **Format:** individual; **Theoretical approach:** unclear | Inpatient settings | – HbA1c; – **Diabetes complications;** – Health service utilisation and healthcare costs; – Psychosocial outcomes; – Diabetes knowledge; patient self-care behaviours; – Physical measures; metabolic | Meta-analysis |
| Alam, 2009[30]; *Patient Education and Counselling* | Glycaemic control and psychological status | **Number of studies:** 35 trials; **Types of studies:** RCTs; **Total sample:** 1431 patients | Psycho-educational interventions | √ BEHA (-); √ DIET; □ DR; □ EXERCISE; □ GC; □ MED; √ PSY (-); □ SMOKING; □ SELF | Generalists; psychological specialists; or did not report the specialist | **Number of session:** 1–16; **Total contact hours:** 20min–28hours; **Duration:** about 13.7 (±11.06) weeks | **Strategies:** face to face; telephone calls; **Format:** group format; a single format and used a combination; **Theoretical approach:** TTM; motivational interviewing | Inpatient settings, other | ↓ **HbA1c;** ↓ **Psychological distress** | Meta-analysis |
| Khunti, 2008[58]; *Diabetic Medicine* | Knowledge and biomedical outcomes | **Number of studies:** nine studies; **Types of studies:** RCTs and RCT was followed by a before-and-after study; **Total sample:** 1004 patients | Any educational intervention | □ BEHA; √ DIET; □ DR; □ EXERCISE; □ GC; □ MED; □ PSY; □ SMOKING; □ SELF | Unclear, did not describe | **Number of session:** unclear; **Total contact hours:** unclear; **Duration:** 3–12 months | **Strategies:** face-to-face; **Format:** group and individual; **Theoretical approach:** unclear | Unclear, did not describe | – Knowledge; – Psychological and biomedical outcome measures | Unclear |
| Loveman, 2008[60]; *Health Technology Assessment* | Clinical effectiveness. | **Number of studies:** 21 published trials; **Types of studies:** RCTs and CCTs; **Total sample:** unclear | Educational interventions | √ BEHA (++); √ DIET (+++); □ DR; √ EXERCISE (+++); √ GC (+++); □ MED; □ PSY; □ SMOKING; √ SELF (+++) | Community workers; diabetes research technician; diabetes nurse, dieticians; educationalist; medical students; nurses; pharmacists; physician or physician assistant | **Number of session:** two to four intensive education of 1.5–2hours followed-up with additional education at, 3 and 6 months; **Total contact hours and duration:** about 150 mins over 6months or 61–52hours over 1year | **Strategies:** face-to-face; **Format:** group and individual; **Theoretical approach:** cognitive-behavioural strategies; pedagogical principle | Inpatient settings, postdischarge, other | – **Diabetic control outcomes;** – Diabetic end points; – QoL and cognitive measures | Narrative review |
| Wens, 2008[78]; *Diabetes Research and Clinical Practice* | Improving adherence to medical treatment recommendations | **Number of studies:** eight studies; **Types of studies:** RCTs and controlling before and after studies; **Total sample:** 772 patients | Interventions aimed at improving adherence to medical treatment | √ BEHA (-); √ DIET (-); □ DR; √ EXERCISE (-); √ GC (-); √ MED (-); □ PSY; □ SMOKING; √ SELF (-) | Diabetes educator; nurse or did not describe | **Number of session:** unclear; **Total contact hours:** unclear; **Duration:** ~9 months or unclear | **Strategies:** face-to-face; telephone; **Format:** face-to-face; group based and telemedicine; **Theoretical approach:** unclear | Inpatient settings, postdischarge, other | – **Adherence;** – HbA1c; – Blood glucose | Cochrane Review Manager software |
| Hawthorne, 2008[53]; *The Cochrane Library* | HbA1c level, knowledge and clinical outcomes | **Number of studies:** a total of 11 trials; **Types of studies:** RCTs; **Total sample:** 1603 patients | Culturally appropriate (or adapted) health education | √ BEHA (-); √ DIET (-); □ DR; √ EXERCISE (-); √ GC (-); □ MED; □ PSY; √ SMOKING (-); □ SELF | Dieticians, diabetes nurses, exercise physiologists; link workers; podiatrists; psychologist and and non-professional link worker | **Number of session:** unclear; **Total contact hours:** unclear; **Duration:** 1 session to 12 months | **Strategies:** face-to-face; booklet; **Format:** group intervention method; one-to-one interviews; mixture of the two methods; purely interactive patient-centred method; semi-structured didactic format and combination of the two approaches; **Theoretical approach:** SAT; Empowerment Behaviour Change Model; Behaviour Change Theory; SCT; Management Model and the Theory of Planned Behaviour | Inpatient settings, postdischarge, other | ↓HbA1c; ↑ **Knowledge scores;** – Other outcome measures | Narrative presentation and meta-analysis |

Continued

**Table 3** Continued

| First author, year; journal | Primary objectives (to assess effect of interventions on…) | Intervention — Educational content | Intervention types | Studies details | Provider | Number of session(s), delivery mode, time, setting | Setting | Outcomes (primary outcomes were in bold; '-': No change; '↑': Increase; '↓': Decrease) | Synthesis methods |
|---|---|---|---|---|---|---|---|---|---|
| Nield, 2007[63]; The Cochrane Library | Metabolic control | Dietary advice | □ BEHA √ DIET □ DR □ EXERCISE □ GC □ MED □ PSY □ SMOKING □ SELF | **Number of studies:** 36 articles (18 trials); **Types of studies:** RCTs; **Total sample:** 1467 participants | Exercise physiologist; dietitian; group facilitator; nutritionist; nurse educator; and physician | **Strategies:** face-to-face; **Format:** group and individual; **Theoretical approach:** unclear — **Number of session:** 1–12; **Total contact hours:** 20min–22 hours; **Duration:** 11 weeks–6 months or unclear | Inpatient settings, postdischarge, other | – **Weight**; – **Diabetic complications**; – **HbA1c**; – QoL; – Medication use; – Cardiovascular disease risk | Meta-analysis |
| Zabaleta, 2007[79]; British Journal of Community Nursing | Clinical effectiveness | Structured group diabetes education | √ BEHA (-) √ DIET (-) □ DR √ EXERCISE (-) √ GC (-) □ MED √ PSY (-) □ SMOKING □ SELF | **Number of studies:** 21 studies; **Types of studies:** controlled trials; **Total sample:** unclear | Diabetes nurse educator; physician's assistant and physicians | **Strategies:** face-to-face; **Format:** group; **Theoretical approach:** unclear — **Number of session:** 4–6 or unclear; **Total contact hours:** 6–12 hours or unclear; **Duration:** 1–6 months or unclear | Postdischarge | –HbA1c | A tabulative synthesis |
| Deakin, 2005[43]; The Cochrane Library | Clinical, lifestyle and psychosocial outcomes | Group-based educational programmes | Did not describe the content of the intervention | **Number of studies:** 14 publications, reporting 11 studies; **Types of studies:** RCTs, and CCTs; **Total sample:** 1532 participants. | Health professionals, lay health advisors | **Strategies:** unclear; **Format:** group; **Theoretical approach:** the Diabetes Treatment and Teaching Programme (DTTP); empowerment model; adult learning model, public health model, HBM and TTM — **Number of session:** unclear; **Total contact hours:** from 6 to 52 hours; **Duration:** 3 hours per year for 2 years and 3 or 4 hours per year for 4 years | Inpatient settings, postdischarge | ↓**Metabolic control**; ↑**Diabetes knowledge**; ↑Empowerment/self-efficacy | Summarised statistically |
| Vermeire, 2005[80]; The Cochrane Library | Improving adherence to treatment recommendations | Interventions that were aimed at improving the adherence to treatment recommendations | □ BEHA □ DIET □ DR □ EXERCISE □ GC □ MED □ PSY □ SMOKING □ SELF | **Number of studies:** 21 articles; **Types of studies:** cross-over study; controlled trial; controlled before and after studies; **Total sample:** 4135 patients | Nurse, pharmacist and other healthcare professionals | **Strategies:** face-to-face; telephone; home visit; video; mailed educational materials; **Format:** unclear; **Theoretical approach:** unclear — **Number of session:** unclear; **Total contact hours:** unclear; **Duration:** unclear | Inpatient settings, postdischarge | **Direct indicators**, such as ↓Blood glucose level; – Indirect indicators, such as pill counts; –Health outcomes | A descriptive review and subgroup meta-analysis |
| Gary, 2003[49]; Diabetes Educator | Body weight and glycaemic control | Educational and behavioural component interventions | □ BEHA √ DIET (-) □ DR √ EXERCISE (-) √ GC (-) √ MED (-) □ PSY □ SMOKING □ SELF | **Number of studies:** 63 RCTs; **Types of studies:** RCTs; **Total sample:** 2720 patients | Nurse (39%); dietitian (26%); physician (17%); other or not specified (23%); exercise psychologist (9%) and health educator (4%) | **Strategies:** unclear; **Format:** unclear; **Theoretical approach:** SAT, contracting model and patient empowerment — **Number of session:** unclear; **Total contact hours:** unclear; **Duration:** 1 month to 19.2 months | Inpatient settings, postdischarge | – **Glycaemic control**; – Weight | Sufficient data were combined using meta-analysis |
| Norris, 2002[65]; Diabetes Care | Total GHb | Self-management education | √ BEHA (-) √ DIET (-) □ DR □ EXERCISE □ GC □ MED □ PSY □ SMOKING √ SELF (-) | **Number of studies:** 31 studies; **Types of studies:** RCTs; **Total sample:** 4263 patients | Dietitian; lay healthcare worker; nurse; physician with team; self (eg, computer-assisted instruction) and team (nurse, dietitian, etc) | **Strategies:** online/web-based; video; face-to-face; phone contact; **Format:** group; individual and mixed; **Theoretical approach:** unclear — **Number of session:** 6 (1–36); **Total contact hours:** 9.2 (1–28) hours; **Duration:** 6 (1.0–27) months | Inpatient settings, post discharge, other | ↓**Total GHb** | Meta-analysis and meta-regression |
| Norris, 2001[64]; Diabetes Care | Clinical outcomes, knowledge, metabolic control | Self-management training interventions | √ BEHA (-) √ DIET (-) □ DR □ EXERCISE □ GC □ MED □ PSY □ SMOKING √ SELF (-) | **Number of studies:** 72 studies (84 papers); **Types of studies:** RCTs; **Total sample:** unclear | CHWs; nurse; or other healthcare professionals | **Strategies:** online/web-based; video (2%); face-to-face; phone contact; **Format:** group; individual and mixed; **Theoretical approach:** SAT; Fishbein and Ajzen HBM — **Number of session:** 1–16; **Total contact hours:** ~22 hours; **Duration:** ~26 months | Inpatient settings, postdischarge, other | ↑**Knowledge**; ↑**Lifestyle behaviours**; –**Psychological and QoL outcomes**; ↑ Glycaemic control; – Cardiovascular disease risk factors | Outcomes are summarised in a qualitative fashion |

Continued

## Table 3 Continued

| First author, year; journal | Primary objectives (to assess effect of interventions on...) | Studies details | Intervention | | | Outcomes (primary outcomes were in bold) | Synthesis methods |
| --- | --- | --- | --- | --- | --- | --- | --- |
| | | | Educational content | Provider | Number of session(s), delivery mode, time, setting | | |

ASE, attitude social influence-efficacy; BCTs, behavioural change techniques; BEHA, behavioural charge (including lifestyle modification); BMI, body mass index; BP, bloodpressure; CCTS, controlled clinical trials; CHD, coronary heart disease; CHW, community health worker; CVR, cardiovascular factors; CVRF, cardiovascular risk factors; DIET, diet; DR, diabetes risks; DSM, diabetes self-management; DSME, diabetes self-management education; EDU, patient education; EXERCISE, exercise; GC, glycaemic regulation; GP, general practice; HbA1c, glycated haemoglobin; HBM, health belief model; HRQoL, health-related quality of life; LDL, low-density lipoprotein cholesterol; LDL-c, low-density lipoprotein cholesterol; LEA, lower extremity amputation; MED, medication; MI, myocardial infarction; PA, physical activity; PRIDE, Problem Identification, Researching one's routine, Identifying a management goal, Developing a plan to reach it, Expressing one's reactions and Establishing rewards for making progress; PSY, psychosocial issues (depression, anxiety); QoL, quality of life; RCTs, randomised controlled trials; SAT, social action theory; SBP, systolic blood pressure; DBP, diastolic blood pressure; HDL–c, high-density lipoprotein cholesterol; SCT, social cognitive theory; SELF, self-management (including problems solving); SMOKING, smoking cessation; SMS, short message system; T2DM, type two diabetes mellitus; TTM, transtheoretical model.

In the educational content: '+': minor focus; '++':moderate focus; '+++' major focus; '-' =unclear what the intensity of the education was for any topic.

In the outcomes: arrow up (↑') for improvement, arrow down (↓') for reduction; a dash ('–') for no change or inconclusive evidence.

### Acute coronary syndrome

The educational content for patients with ACS covered cardiovascular risk factors in eight reviews (53.33%), psychosocial issues in eight reviews (53.33%), smoking cessation in six reviews (40.00%), exercise in five reviews (33.33%), behavioural change in five reviews (33.33%), diet in four reviews (26.67%), self-management in three reviews (20.00%) and medication in one review (6.67%). Two reviews only included smoking cessation and cardiovascular risk factors. The most common educational providers were nurses and a multidisciplinary team. Six studies[31 36 48 51 56 69] (6/15, 40%) described the theoretical approach that underpinned the education intervention.

### Type 2 diabetes mellitus

The educational content for patients with T2DM included diet in 23 reviews (63.89%), behavioural change in 21 reviews (58.33%), self-management in 20 reviews (55.56%), exercise in 17 reviews (47.22%), glycaemic regulation in 16 reviews (44.45%), medication in 13 reviews (36.11%), psychosocial issues in 9 reviews (25.00%), smoking cessation in 2 reviews (5.56%), cardiovascular risk factors in 2 reviews (5.56%) and DM risks in 1 review (2.78%). The most common providers were dieticians, nurses and a multidisciplinary team. The number of sessions, total contact hours and durations varied. Thirteen reviews[30 33 43 49 52–54 60 64 67 75–77] (13/36, 36.11%) described the theoretical approach that underpinned the education intervention.

### Effect of interventions

The outcomes of the included systematic reviews and meta-analyses are summarised in table 4.

### Patients with ACS

Three major types of health education-related interventions were used for patients with ACS: general health education (only included general health information), psychoeducational interventions and secondary prevention educational interventions (including strategies to promote a healthy lifestyle, manage medications and reduce cardiovascular complications) as well as internet-based interventions.

#### General health education

The findings are based on our synthesis of the findings from six systematic reviews.[37 48 50 51 59 70] Overall, there were mixed effects of general health education on behavioural change or clinical outcomes in patients with ACS. There was *some evidence* of a positive effect of general health education on knowledge, behaviour, psychosocial indicators, beliefs and risk factor modification, but no effects for key clinical outcomes, such as cholesterol level, hospitalisation, mortality, MI and revascularisation. The results for health-related quality of life, healthcare utilisation and costs were mixed; several reviews reported a significant change, and other reviews reported no significant change for these outcomes. Only one review focused on telephone-based health education. There is *some evidence* that

**Table 4** Summary of evidence from quantitative research syntheses

| Intervention | Number of systematic reviews/meta-analysis, total participants | First author, year | Primary results/findings | | Rating the evidence of effectiveness |
|---|---|---|---|---|---|
| **Patients with acute coronary syndrome** | | | | | |
| General health education | Six/161 997 patients (Goulding et al, 2010[51] did not give the total sample size) | Ghisi, 2014[50] | Knowledge | 91% studies* | Some evidence |
| | | | Behaviour | 77%/84%/65% studies* | |
| | | | Psychosocial indicators | 43% studies* | |
| | | Brown, 2013[37] | Mortality | | |
| | | | MI | | |
| | | | Revascularisations | | |
| | | | Hospitalisations | | |
| | | | HRQoL | | |
| | | | Withdrawals/dropouts | | |
| | | | Healthcare utilisation and costs | | |
| | | Brown, 2011[70] | Total mortality | | |
| | | | MI | | |
| | | | CABG | | |
| | | | Hospitalisations | | |
| | | | HRQoL | 63.6% studies* | |
| | | | Healthcare costs | 40% studies* | |
| | | | Withdrawal/dropout | | |
| | | Goulding, 2010[51] | Beliefs | 30.08% studies* | |
| | | | Secondary outcomes | | |
| | | Fernandez, 2007[48] | Smoking | | |
| | | | Cholesterol level | | |
| | | | Multiple risk factor modification | | |
| | | Kotb, 2014[59] | All-cause hospitalisation | | |
| | | | All-cause mortality | | |
| | | | Smoking cessation | | |
| | | | Depression | | |
| | | | Systolic blood pressure | | |
| | | | Low-density lipoprotein | | |
| | | | Anxiety | | |
| Psychoeducational interventions | Six/37 883 patients | Barth, 2015[69] | Abstinence by self-report or validated | | Sufficient evidence |
| | | Dickens, 2013[45] | Depression | | |
| | | Aldcroft, 2011[31] | Smoking cessation | | |
| | | | Physical activity | | |
| | | Huttunen-Lenz,2010[56] | Prevalent smoking cessation | | |
| | | | Continuous smoking cessation | | |
| | | | Total mortality | | |
| | | Barth, 2008[36] | Abstinence by self-report or validated | | |
| | | | Smoking status | | |
| | | Barth, 2006[35] | Abstinence | | |
| | | | Smoking status | | |

Continued

**Table 4** Continued

| Intervention | Number of systematic reviews/meta-analysis, total participants | First author, year | Primary results/findings | | Rating the evidence of effectiveness |
|---|---|---|---|---|---|
| Secondary prevention educational interventions (including Internet-based secondary prevention) | Three/25 154 patients | Devi, 2015[44] | Mortality | | Some evidence |
| | | | Revascularisation | | |
| | | | Total cholesterol | | |
| | | | HDL cholesterol | | |
| | | | Triglycerides | | |
| | | | HRQOL | | |
| | | Auer, 2008[34] | All-cause mortality | | |
| | | | Readmission rates | | |
| | | | Reinfarction rates | | |
| | | | Smoking cessation rates | | |
| | | Clark, 2005[41] | Mortality | | |
| | | | MI | | |
| | | | Quality of life | Most of the included studies* | |
| **Patients with T2DM** | | | | | |
| General health education | Five/2319 patients (Choi et al, 2016[40]; Loveman et al, 2008[60]; Zabaleta et al, 2007[79] did not give the total sample size) | Choi, 2016[40] | HbA1c | | Some evidence |
| | | Saffari, 2014[74] | Glycaemic control | | |
| | | Duke, 2009[46] | HbA1c | | |
| | | | BP | | |
| | | | Knowledge, psychosocial outcomes and smoking habits | No data | |
| | | | Diabetes complications or health service utilisation and cost analysis | No data | |
| | | Loveman, 2008[60] | Diabetic control outcomes | 46.15% studies* | |
| | | | Weight | 66.67% studies* | |
| | | | Cholesterol or triglycerides | 40.00% studies (+) | |
| | | Zabaleta, 2007[79] | HbA1c | 4.8% studies* | |
| Culturally appropriate health education | Eight/20 622 patients (Ricci-Cabello et al, 2014[73] and Gucciardi et al, 2013[52] did not give the total sample size) | Creamer, 2016[42] | HbA1c | | Some evidence |
| | | | HRQoL | | |
| | | | AEs | No AEs | |
| | | Ricci-Cabello, 2014[73] | HbA1c | | |
| | | | Diabetes knowledge | 73.3% studies* | |
| | | | Behaviours | 75% studies* | |
| | | | Clinical outcomes | Fasting blood glucose, HbA1c and BP improved in 71%, 59% and 57% of the studies | |

Continued

**Table 4** Continued

| Intervention | Number of systematic reviews/meta-analysis, total participants | First author, year | Primary results/findings | | Rating the evidence of effectiveness |
|---|---|---|---|---|---|
| | | Attridge, 2014[33] | HbA1c | | |
| | | | Knowledge scores | | |
| | | | Clinical outcomes | | |
| | | | Other outcome measures | Showed neutral effects | |
| | | Gucciardi, 2013[52] | HbA1c levels | 3 of 10 studies* | |
| | | | Anthropometrics | 3 of 11 studies* | |
| | | | Physical activity | One of five studies* | |
| | | | Diet outcomes | Two of six studies* | |
| | | Nam, 2012[62] | HbA1c level | | |
| | | Hawthorne, 2010[54] | HbA1c | | |
| | | | Knowledge scores | | |
| | | Khunti, 2008[58] | Knowledge levels | Only one study reporting a significant improvement | |
| | | | Biomedical outcomes | Only one study reporting a significant improvement | |
| | | Hawthorne, 2008[53] | HbA1c | | |
| | | | Knowledge scores | | |
| | | | Other outcome measures | | |
| Lifestyle interventions+ behavioural programme | Six/10 440 patients (Huang et al, 2016[55]; Pillay et al, 2015[71] and Ramadas et al, 2011[77] did not give the total sample size) | Huang, 2016[55] | HbA1c | | Some evidence |
| | | | BMI | | |
| | | | LDL-c and HDL-c | | |
| | | Chen, 2015[39] | HbA1c | | |
| | | | BMI | | |
| | | | SBP | | |
| | | | DBP | | |
| | | | HDL-c | | |
| | | Terranova, 2015[72] | HbA1c level | | |
| | | | Weight | | |
| | | Pillay, 2015[71] | HbA1c levels | | |
| | | | BMI | | |
| | | Ramadas, 2011[77] | HbA1c | 46.2% studies * | |
| | | Gary, 2003[49] | Fast blood sugar | | |
| | | | Glycohaemoglobin | | |
| | | | HbA1 | | |
| | | | HbA1c | | |
| | | | Weight | | |

**Table 4** Continued

| Intervention | Number of systematic reviews/meta-analysis, total participants | First author, year | Primary results/findings | | Rating the evidence of effectiveness |
|---|---|---|---|---|---|
| Self-management educational interventions | Nine/19 597 patients (Minet et al, 2010[61]; Fan et al, 2009[47] and Norris et al, 2001[64] did not give the total sample size) | Pal, 2014[67] | Cardiovascular risk factors | | Sufficient evidence |
| | | | Cognitive outcomes | | |
| | | | Behavioural outcomes | Only one study reporting a significant improvement | |
| | | | AEs | No AEs | |
| | | Vugt , 2013[75] | Health behaviours | 7 of 13 studies * | |
| | | | Clinical outcomes measures | Nine studies * | |
| | | | Psychological outcomes | Nine studies * | |
| | | Pal , 2013[68] | HbA1c | | |
| | | | Depression | | |
| | | | Quality of life | | |
| | | | Weight | | |
| | | Steinsbekk, 2012[76] | HbA1c | | |
| | | | Main lifestyle outcomes | | |
| | | | Main psychosocial outcomes | | |
| | | Minet, 2010[61] | Glycaemic control | | |
| | | Fan, 2009[47] | Diabetes knowledge | | |
| | | | Overall self-management behaviours | | |
| | | | Overall metabolic outcomes | | |
| | | | Overall weighted mean effect sizes | | |
| | | Deakin, 2005[43] | Metabolic control (HbA1c) | | |
| | | | Fasting blood glucose levels | | |
| | | | Weight | | |
| | | | Diabetes knowledge | | |
| | | | SBP | | |
| | | | Diabetes medication | | |
| | | Norris, 2002[65] | Total GHb | | |
| | | Norris, 2001[64] | Knowledge | | |
| | | | Self-monitoring of blood glucose | | |
| | | | Self-reported dietary habits | | |
| | | | Glycaemic control | | |
| Therapeutic education | One/total sample: unclear | Odnoletkova, 2014[66] | Cost-effectiveness | Overall high in studies on prediabetes and varied in studies on T2DM | Insufficient evidence |
| Foot health education | One/total sample: unclear | Amaeshi[32] | Diabetes complications | | Some evidence |
| | | | Incidence of LEA | | |

Continued

**Table 4** Continued

| Intervention | Number of systematic reviews/meta-analysis, total participants | First author, year | Primary results/findings | | Rating the evidence of effectiveness |
|---|---|---|---|---|---|
| Group medical visit | One/2240 patients | Burke, 2011[38] | HbA1c | | Some evidence |
| | | | BP and DBP | | |
| | | | SBP | | |
| | | | Cholesterol—LDL | | |
| Psychoeducational intervention | One/1431 patients | Alam, 2009[30] | HbA1c | | Some evidence |
| | | | Psychological status | | |
| Interventions aimed at improving adherence to medical treatment recommendations | Three/4907 patients (Lun Gan et al, 2011[57] did not give the total sample size) | Lun Gan, 2011[57] | Oral hypoglycaemic adherence | Five of seven studies * | Some evidence |
| | | Wens et al., 2008[78] | Adherence | General conclusions could not be drawn | |
| | | Vermeire, 2005[80] | HbA1c | | |
| Dietary advice | One/1467 patients | Nield, 2007[63] | Glycaemic control (addition of exercise to dietary advice) | | Insufficient evidence to determine |
| | | | Weight | Limited data | |
| | | | Diabetic microvascular and macrovascular diseases | Limited data | |

*Intervention group is significantly better than control group, for example, '91% studies ' means 91% studies reported a significant better compared with control group.

AEs, adverse events; BMI, body mass index; BP, blood pressure; CABG, coronary artery bypass graft surgery; HbA1c, glycated haemoglobin; HRQoL, health related quality of life; LDL-c, low-density lipoprotein cholesterol; LEA, lower extremity amputation; MI, myocardial infarction; RCTs, randomised controlled trials; SBP, systolic blood pressure, DBP, diastolic blood pressure, HDL-c, high density lipoprotein cholesterol; T2DM, type two diabetes mellitus.

telephone-based health education during cardiac rehabilitation might improve all-cause hospitalisation, anxiety, depression, smoking cessation and systolic BP, but there is no evidence for improvements in all-cause mortality and reductions in low-density lipoprotein cholesterol.[59]

Psychoeducational interventions

Strategies for psychoeducational interventions have a specific focus on smoking cessation and depression. The findings are based on synthesis of results from six publications.[31 35 36 45 56 69] There is *sufficient evidence* that psychoeducational programmes are effective at decreasing smoking, achieving smoking abstinence and reducing depression. One review reported no effect on smoking cessation[31] or total mortality.[56]

Secondary prevention educational interventions

The following statements are based on our synthesis of results from three papers.[34 41 44] There is *some evidence* that secondary prevention educational interventions reduce MI readmission rates and improve quality of life, but the intervention was ineffective in reducing revascularisation, cholesterol levels and improving smoking cessation rates. The results are mixed for mortality and re-infarction rates; two reviews[34 41] found positive effects on mortality, while one review[44] did not.

Patients with T2DM

Ten types of health education-related interventions were used for patients with T2DM: culturally appropriate health education (tailored to the religious beliefs, culture, literacy and linguistics of the geographical area), dietary advice, foot health education, group medical visits (a group education component taught by health professionals), general health education (only included general health information), improving the uptake and maintenance of medication regimes (eg, promoting the use of oral hypoglycaemic medications), lifestyle interventions (specific focus on dietary changes and increased physical activity, or stress management), psychoeducational interventions and self-management educational interventions (activities that promote or maintain the behaviours to manage T2DM often based on the National Standards for Diabetes Self-Management Education[13]) and therapeutic education (collaborative process needed to modify behaviour and more effectively manage risk factors).

Culturally appropriate health education

Findings are based on our synthesis of results from eight publications.[33 42 52–54 58 62 72] Overall, there was *some evidence* of the effects of culturally appropriate health education on clinical outcomes for T2DM. There was *sufficient*

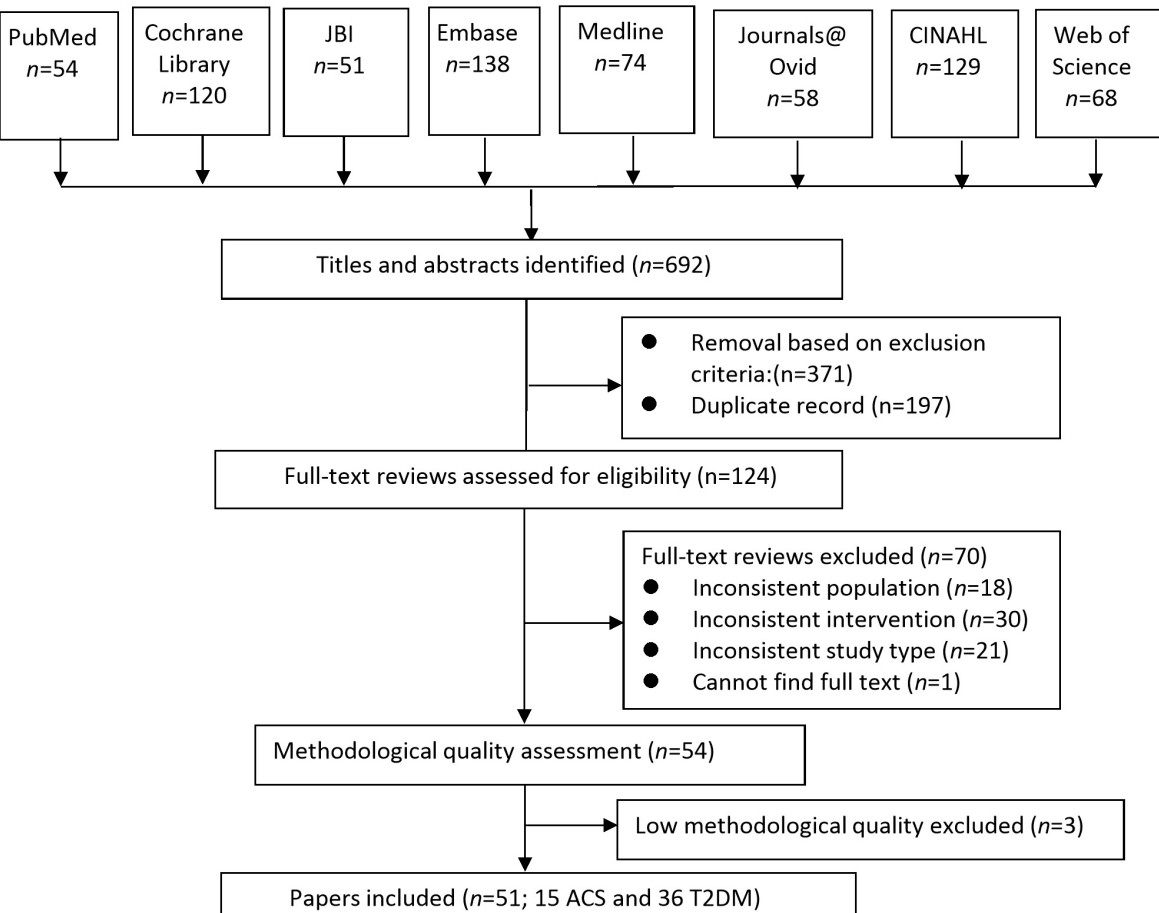

**Figure 1** Flow chart of the systematic reviews and meta-analyses selection process.

evidence that culturally appropriate health education improves HbA1c reduction and knowledge scores. There is *some* evidence that physical activity and clinical outcomes (blood glucose, HbA1c, BP) were improved. There were no data relating to adverse events during the intervention and follow-up (such as hypoglycaemic events and mortality), and there was insufficient evidence about improvements in quality of life.

### General health education
The statements are based on our synthesis of results from five papers.[40 46 60 74 79] Overall, there were mixed effects of general health education programmes on clinical outcomes for T2DM, including HbA1c, cholesterol level and triglyceride level. There was *some evidence* of the effectiveness of general health education on the management of glycaemia, weight reduction and some diabetes management outcomes (HbA1c, diabetes complications). There were no data supporting the effectiveness of general health education on reduced health service utilisation, diabetes complications, improved knowledge, psychosocial outcomes or smoking habits.

### Lifestyle interventions
The following statements are based on our synthesis of results from six reviews.[39 49 55 71 72 77] Overall, there were mixed effects of the lifestyle interventions on cholesterol

level, HbA1c level and body weight. There is *some evidence* that lifestyle interventions or behavioural programmes are effective for blood glucose and BP management, but they were ineffective for reductions in HbA1c scores.[71 72]

### Uptake and maintenance of medication regimes
The statements are based on our synthesis of results from three publications.[57 78 80] There is *some evidence* of the effectiveness of increased uptake and maintenance of medication regimes for taking medications for HbA1c regulation including oral hypoglycaemic agents.

### Self-Management educational interventions
The statements are based on our synthesis of results from nine reviews.[43 47 61 64 65 67 68 75 76] Overall, there was *sufficient evidence* of the effects of self-management education interventions on HbA1c level, knowledge, lifestyle outcomes and main psychosocial outcomes. However, there was *insufficient* evidence of the benefits of this education intervention on depression, quality of life and body weight.

### Other health education-related interventions
Other health education-related interventions for patients with T2DM included therapeutic education, foot health education, group medical visits, psychoeducational interventions and dietary advice. Statements for all of these

interventions are based on our synthesis of results from one review.

There is *some evidence* that foot health education is effective in reducing the incidence of lower extremity amputation.[32] There is *some evidence* that group medical visits are effective for improving HbA1c and systolic BP management.[38] There is also *some evidence* that psychoeducational programmes are effective for improving HbA1c regulation and psychological status.[30]

Finally, there is *insufficient evidence* that dietary advice improves glycaemic and weight management or reduces microvascular and macrovascular diseases.[63] There is also *insufficient evidence* for the cost-effectiveness of therapeutic education for patients with T2DM.[66]

## DISCUSSION

This umbrella review identified 51 systematic reviews or meta-analyses (15 for ACS and 36 for T2DM) that assessed the outcomes of various aspects (such as the duration, contact hours, educational content, delivery mode) of the delivery of health education-related interventions relevant to high-risk patients with ACS and T2DM. Health education has become an integral part of the management for people with ACS and T2DM. The most appropriate focus of the education provided to patients with ACS and T2DM remains largely undefined in the literature. For example, it remains unknown if the focus should be primarily on cardiovascular risk factors, blood glucose monitoring or all educational components for patients with both conditions.[70 76] In addition, should cardiovascular risk factors be the focus during the acute inpatient stay with other educational needs such as the smoking cessation occurring within the primary care or outpatient settings.[31 69 70]

It remains challenging to determine the specific strategy or format that is the most effective delivery mode for patients with ACS or T2DM. There is very limited evidence to guide clinicians on the duration, contact hours, educational content, delivery mode, total length and setting of health education programme for cardiac patients.[50] For patients with DM, one study reported that more successful programme were longer than 6 months (longer duration), consisted of greater than 10 contact sessions (high intensity) and were one-on-one sessions with individualised assessment.[82]

### Use of theoretical orientation to develop educational intervention
#### For patients with ACS
Use of theory when designing behavioural change interventions may also influence effectiveness.[75] Health education using a cognitive behavioural strategy is most consistently effective in changing maladaptive illness beliefs,[51] and studies using more than two behavioural change strategies reported significant differences between the intervention and control groups.[31] In one review, a significant change in smoking cessation was not observed in subgroup analyses between studies that did or did not report using a theory in intervention planning[56]; however, the authors did not suggest that using a theory in programme planning should be disregarded but reported that examining actual theories or mechanisms underlying health education programmes is required.[56] Owing to the considerable overlap between different theories and the detailed description of the theoretical approach in only approximately 40% of the included papers, it is difficult to determine the most effective theoretical approach, but many models can be used with success, such as the health belief model (HBM), social cognitive theory (SCT) and transtheoretical model (TTM).[56 67 69 75] Three reviews[31 41 44] noted that some included studies used behavioural strategies such as goal setting. These strategies were found to be beneficial for patients with coronary heart disease.

### For patients with T2DM
Although the theoretical approach underpinning the health education programme was not always described, 13 of the 36 reviews (36.11%) related to T2DM reported the theoretical approach used in their included studies. The most common theories were SCT (including self-efficacy), empowerment theories (eg, empowerment behaviour change model, self-determination and autonomy motivation theory, middle-range theory of community empowerment) and TTM. There is evidence that health education interventions based on a theoretical model are likely to be effective.[43] Vugt *et al* suggested that self-care education programmes should be based on theories and that theory-based self-care interventions are more effective than non-theory-based programmes.[75 83] Theories could help to specify the key target health behaviours and behavioural change techniques required to generate the desired outcomes.[75] The decision regarding the theory should be based on the aim of the programme and factor for intervention.[77] Only one review reported that a theoretical approach underpinning the health education programme is not necessary for better outcomes.[76] Fourteen reviews[30 33 40 46 52 57 60 63 64 67 68 73 75 77] reported that goal setting was conducted in the included studies. Goal setting by patients, health professionals or mutually agreed goals were linked to improved patient outcomes.

### Educational content
#### For patients with ACS
Most reviews reported that the educational content of the interventions was comprehensive. The most common topics, of the average 3.7 topics per education session, were behavioural change, cardiovascular risk factors management, exercise, psychosocial issues and smoking cessation. An underlying principle of health education for patients with ACS is that knowledge is necessary, but not enough to develop health behaviours and change risk factors.[31 50] Age, cognitive factors, environmental factors and social and economic background are also important considerations.[50] While interventions using a behavioural programme, telephone-based content or

self-care are effective for smoking cessation, there was insufficient evidence to support that any type of educational programme was more efficacious than the others.[69] Psychoeducation, which is defined as multimodal, educationally based, self-management interventions,[31] led to enhanced physical activity levels within 6–12 months when added to cardiac rehabilitation (CR) and was more effective than an exercise programme or health education alone.[31 56] Moreover, psychoeducational interventions were more effective for patients with ACS than other types of health education.[31 56]

### For patients with T2DM

The educational content for patients with T2DM focused more on behavioural change, diet, exercise, glycaemic regulation, medication and self-management. Health education that was self-management was more effective for patients with T2DM.[40 47] In addition, based on the current evidence, the educational content should be culturally sensitive, especially for patients with T2DM[33 42 54]; culturally appropriate diabetes health education may have a greater impact on the management of glycaemia and reduce diabetes complications.[77] The educational interventions for patients with T2DM focused primarily on HbA1c, lipid levels, quality of life and body weight. HBM and SCT were the most common theories used in the included reviews.

### Teaching strategies and outcomes
#### For patients with ACS

Most reviews reported that the education was provided using multiple teaching methods and in multiple settings. Nurses and multidisciplinary teams were the most frequent people providing education, and most education programmes were delivered postdischarge. Although face-to-face sessions were the most common delivery format, many education sessions were also delivered by telephone or through individualised counselling. Telephone-based health education appeared to be effective for reducing hospitalisations, systolic BP, smoking rates, depression and anxiety.[59] The educational interventions for patients with ACS focused primarily on clinical outcomes (hospitalisation and mortality), modifiable risk factors (BP, low-density lipoprotein levels and smoking cessation) and other psychological outcomes (anxiety and depression).

#### For patients with T2DM

Mixed health education programmes generally included group sessions combined with educator-facilitated individual sessions, covering basic knowledge and problem-solving skills. These programmes produced greater benefits and larger effect sizes for blood glucose reduction and knowledge levels in patients with T2DM.[47] In contrast, individual education programmes have been reported as more effective in achieving outcomes than group-based education. This may be because education programmes might be more efficient at addressing

personal needs, with greater participant engagement.[73] However, one systematic review reported that individual and group patient education demonstrated similar outcomes among patients with T2DM.[46]

Although face-to-face sessions were the most common delivery format, many education sessions were also delivered by telephone or individualised counselling. Face-to-face health education programmes were most effective for enhancing blood glucose regulation and knowledge levels, while mixed delivery models (face-to-face, phone contact, online or web-based or video) produced a moderate effect for knowledge levels.[47] Another review reported that face-to-face health education programmes generated a greater benefit for metabolic management than those delivered using electronic communication technology.[73]

Nurses (including diabetes nurses educators), community workers, dieticians and multidisciplinary teams were the most frequent educators, and most of the education programmes were delivered postdischarge. Some reviews indicated that health education programmes delivered by a group of different educators, with some degree of education reinforcement at additional points of contact, may provide the best results.[60 76] However, based on two studies that reported HbA1c at 12 months, it is indicated that the outcomes in studies with only a diabetes nurse as the educator also tended to do better than the outcomes in studies with a multidisciplinary team, while the biggest effect was seen when a dietician was the only educator.[76] Health education programmes delivered by one person may focus more on the patient's ability than the educational content or quality of the health education programmes.[76] However, no clear conclusion can be drawn whether having one educator delivering the intervention is best due to few information.[60]

### Delivery, timing and follow-up
#### For patients with ACS

Most educational sessions were delivered weekly. Few reviews provided information regarding the duration of education interventions; when the duration was reported, it varied from 4 weeks to 48 months. These findings suggest that there is a significant gap in the evidence in relation to the duration, contact hours, educational content, optimal delivery mode, total length and setting of health education programmes for cardiac patients.[50] For patients with ACS, one systematic review that included 7 studies with a total of 536 participants reported that studies with education lasting at least 6 months resulted in the most significant changes in the primary outcomes (such as behavioural change, smoking cessation)[31] and that at least 12 months of follow-up is needed to evaluate the impact of telephone-based education.[59] Another review reported that the intensity of education programmes is important for efficacy regarding smoking cessation: interventions with a very low intensity and brief interventions do not have a significant effect,[69] and programmes for smoking cessation among patients with coronary heart

disease should last >1 month.[69] Most of the reviews were provided for patients with ACS in inpatient settings and then within postdischarge settings, five reviews[31 36 45 48 59] did not explicitly state the settings in which the health education-related interventions were provided.

### For patients with T2DM

Education sessions were delivered weekly or monthly. Longer health education programmes for T2DM (>6 months) produced larger effects for all primary outcomes (such as HbA1c).[47] Health education lasting >3 months resulted in the largest effect size compared with health education of a shorter duration (<3 months).[33] For HbA1c, the effect size at 6 months seemed to be significantly greater than at 3 and 12 months; in other words, the effect size peaked at 6 months.[62] In general, health education of a greater intensity (longer duration and more sessions) was more effective for blood glucose reduction and knowledge levels among patients with T2DM.[47 74] Compared with health education programmes covering only one topic, programmes that included multiple or mixed educational topics yielded consistently greater benefits in blood glucose reduction and knowledge levels.[47] In addition, health education programmes combined with specific behavioural change strategies (such as self-care strategies) seemed more effective than other programmes.[47] Health education-related

interventions were mainly delivered in hospital settings, primary care settings, diabetes centres or community-based settings, although six reviews[32 39 55 58 67 72] did not explicitly state the delivery settings.

### Recommendations about health education interventions for patients with ACS and T2DM

These results from included systematic reviews and meta-analyses help to provide recommendations about the content of a health education intervention for patients with ACS and T2DM, requiring further evaluation. Future development of educational programmes for patients with ACS and T2DM by healthcare professionals should consider the needs of people with these diseases.[37 40 42 70] Based on the results and findings from this umbrella review, recommendations are made in table 5. The acute life-threatening nature of ACS requires that increased emphasis should be placed on cardiovascular risk factors in any combined education programme. Both ACS and T2DM have common lifestyle factors such as inactivity and high fat diet requiring modifications.

### Overall completeness and applicability of evidence

This overview potentially provides an estimate with the lowest level of bias for the impact of health education-related interventions for patients with ACS or T2DM and could be regarded as an all-inclusive summary of the

**Table 5** Recommendations of health education programmes for patients with ACS and T2DM

| | | Patients with ACS | Patients with T2DM | Both ACS and T2DM |
|---|---|---|---|---|
| Theoretical approach | | SCT, empowerment theories. | HBM; SCT. | HBM; SCT and empowerment theories |
| Behavioural strategies | | Goal setting | Goal setting | Goal setting |
| Educational content | | Behavioural change (such as smoking cessation), cardiovascular risk factors, exercise, medication and psychosocial issues | Behavioural change, diet, exercise, glycaemic control, medication and self-management | Behavioural change (such as smoking cessation), cardiovascular risk factors, diet, exercise, glycaemic control, medication, psychosocial issues and self- management |
| Healthcare professionals to deliver | | Nurse or multidisciplinary team | Multidisciplinary team; dietitian or nurse | Nurse or multidisciplinary team |
| Teaching approaches | Strategies | Face to face; telephone or mixed | Face-to-face, written materials; telephone or mixed | Face-to-face, written materials; telephone contact or mixed |
| | Format | Individual (one by one) or mixed | Individual (one by one) or mixed | Individual (one by one) or mixed |
| Delivery timing | Contact hours | More than 30 min per time per week | More than 30 min per time per week | More than 30 min per time per week |
| | Duration | At least 6 months | About 6 months | At least 6 months |
| Duration of follow-up | | At least 12 months | At least 12 months | At least 12 months |
| Settings | | Inpatient and postdischarge settings | Hospital settings and primary care settings | Inpatient and postdischarge settings |

ACS, acute coronary syndrome; T2DM, type two diabetes mellitus; SCT, social cognitive theory; HBM, health belief model.

current evidence base for health education for these patients. While this umbrella review identified evidence for each of the types of health education, there was only a small number of reviews within some categories (such as psychoeducational intervention[30] and dietary advice[63]), and these studies were not very informative. This umbrella review also found no reviews that systematically analysed varying doses of health education; therefore, could not examine the dose-response effects. There was insufficient information about the evaluated doses (total contact hours and duration of education) to enable comparison of the benefits of differences in the magnitude of the doses across the different research. This umbrella review found no reviews focused on patients with ACS and T2DM; instead, all of the systematic reviews and meta-analyses focused on only one of these diseases.

## Quality of the evidence

The methodological quality of the included systematic reviews and meta-analyses varied. All of the included reviews or meta-analyses were of moderate-to-high methodological quality, as assessed using AMSTAR. However, only 30 (58.8%) systematic reviews or meta-analyses were rated as high quality and only 3 (5.9%) systematic reviews or meta-analyses[43 53 69] adequately met all 11 AMSTAR criteria. This indicates that some of the reviews included in this umbrella review may have limitations in their design, conduct and/or reporting that could have influenced the findings when considered both individually and collectively.[32 65]

The quality of the primary studies in the included systematic reviews or meta-analyses also varied. The main sources of bias were inadequate reporting of allocation concealment and randomisation processes, as well as lack of outcome blinding.[33 42 69 70] This bias in the methodological quality led to lower quality assessments, which varied by results within each included review. Other reasons for lower methodological quality included heterogeneity in, or inconsistency of, the effect and imprecise findings. Heterogeneity between studies in this umbrella review was described in terms of the intervention, participant characteristics and length of follow-up. Heterogeneity was an important factor indicating the complexity of the health education interventions.[56] The variability in the approaches, tools or scales used to measure outcomes between the included studies are likely to introduce some heterogeneity.[30] The heterogeneity of the educational interventions seen in the reviews included in this umbrella review may reflect the uncertainty about the optimal strategy for providing health education to patients.[37] In addition, 240 studies were included more than once in the included reviews and meta-analyses. However, the overall overlap of studies among reviews and meta-analyses-related ACS and T2DM was slight, CCA of 2.6% and 2.1%, respectively.[25]

This umbrella review is the first synthesis of systematic reviews or meta-analyses to take a broad perspective on health education-related interventions for patients with ACS or T2DM. Given that health education is complex, the biggest challenge for systematic reviews or meta-analyses of health education is accounting for the potential clinical heterogeneity in health education-related interventions (content and delivery approaches) and the population of patients who receive health education. To facilitate comparisons across systematic reviews of health education and the efficient future update of this umbrella review, future reviews or meta-analyses need high-quality research and to standardise their design and reporting, including the reporting of included study characteristics, assessment criteria for risk of bias, outcomes and methods to synthesise evidence synthesis.

## CONCLUSIONS

For clinicians providing educational interventions to individuals with ACS and T2DM, the results from this review provide a contemporaneous perspective on current evidence on the effectiveness of health education (its content and delivery methods) for this high-risk patient group. The current evidence compiled by this umbrella review supports current international clinical guidelines, that theoretically based education interventions lasting 6 months, delivered in multiple modes (face to face, phone contact, online or web-based or video), and with individualised education delivered weekly, are more likely to generate positive outcomes. This review also supports health education-related interventions provided by health professionals, including nurses and multidisciplinary teams, delivering content including specific clinical factors for ACS and T2DM (BP, glycaemic level and medication), modifiable risk factors (unhealthy diet, inactivity and smoking) and other psychological factors (anxiety and depression). These health education interventions could be delivered postdischarge, such as rehabilitation centres, primary care centres and the community and should be at least 6 months in duration. The effectiveness of these programmes was based on HbA1c levels, knowledge, psychosocial outcomes, readmission rates and smoking status rather than clear evidence of reduced mortality, MI or short-term and long-term complications. In addition, psychoeducational interventions were more effective for patients with ACS, and health education that was culturally appropriate or taught self-management was more effective for patients with T2DM. We also found that longer durations and high-intensity health education provided in an individualised format were more helpful for patients with ACS or T2DM.

The fact that none of the included reviews included patients with both ACS and T2DM indicates a clear need for further rigorous experimental studies with patients with both diseases. Future research that includes these aspects of education are likely to determine the effectiveness of educational interventions focusing on cardiovascular and DM risk factors and complications within patients with ACS and T2DM.

**Author affiliations**
[1]Tenth People's Hospital, Tongji University, Shanghai, China
[2]School of Nursing, Midwifery and Paramedicine, Australian Catholic University, Brisbane, QLD, Australia
[3]School of Nursing, Jinggangshan University, Ji'An, China
[4]Melbourne Health, La Trobe University, Melbourne, Victoria, Australia
[5]School of Nursing, Midwifery and Paramedicine, University of the Sunshine Coast, Australia
[6]School of Nursing, Queensland University of Technology (QUT), Brisbane, Australia
[7]Royal Brisbane and Women's Hospital (RBWH), Australia
[8]Mater Medical Research Institute-University of Queensland (MMRI-UQ), Australia
[9]Faculty of Health Sciences, Australian Catholic University, North Sydney, NSW, Australia
[10]Ingham Institute of Applied Medical Research, Sydney, NSW, Australia

**Acknowledgements** We would like to thank the authors of the original articles who provided additional unpublished data.

**Contributors** Study conception and design: XL-L, MJ, KW, C-JW, YS. Data collection: XL-L, YS. Data analysis: XL-L, YS, MJ, KW, C-JW. Manuscript drafts: XL-L, MJ, C-JW, KW, YS.

**Funding** This research received no specific grant from any funding agency in the public, commercial or not-for-profit sectors. The lead author is a recipient of an Australian Catholic University Faculty of Health Sciences Tongji University Cotutelle PhD Scholarship.

**Competing interests** None declared.

**Provenance and peer review** Not commissioned; externally peer reviewed.

**Data sharing statement** No additional data are available.

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
