## [Reviewer comments · BMJ Open]

ARTICLE DETAILS

TITLE (PROVISIONAL)	Health Education for Patients with Acute Coronary Syndrome and Type 2 Diabetes Mellitus: An Umbrella Review of Systematic Reviews and Meta-Analyses
AUTHORS	Liu, Xian-liang; Shi, Yan; Willis, Karen; Wu, Jo; Johnson, Maree

VERSION 1 - REVIEW

REVIEWER	Martha Funnell, MS, RN, CDE, FAAN University of Michigan US
REVIEW RETURNED	02-May-2017

GENERAL COMMENTS	This well-written manuscript addresses an area that is of great interest to professionals who provide care and education for people with ACS and T2D. The umbrella methodology is clearly described, innovative and comprehensive. This manuscript provides a systematic approach to this topic that could be valuable to clinicians and researchers. Issues to address: • It was not clear that the practical differences between T2D and ACS education were appreciated and how these differences influence the study methods. For example, very few people with T2D are diagnosed in the hospital or are referred to education after “discharge”. The vast majority of T2D programs are offered in outpatient facilities. In addition, many of the recent studies were designed to meet national or international education standards (e.g., American Diabetes Assn, International Diabetes Assn) and payer requirements. This needs to be referred to understand the context of the reviews.• On page 7 in the Intervention Type section: List the health professionals in the order of frequency of delivery. Nurses (including diabetes nurses educators) are later noted as most often providers, not community healthcare workers, doctors etc. as stated.• Please clarify that “no evidence” includes only measured outcomes. Also, define “self-management in 20 reviews” in T2D. Could these be considered standardized diabetes education programs that meet national or international guidelines?• Pages 20 and 21: Please state how effectiveness is defined in “appears to ineffective for HA1C”. In addition, improving adherence to medical treatment is very vague. Do you mean medication taking? The word “diabetic” is not used to refer to either people with diabetes or outcomes. Several organizations (included Australia Diabetes Educators) have recent position statements on this issue.• Were more recent psychosocial measures than of quality of life, such as disease-related distress assessed in the articles reviewed?• Although one study for T2D reported that one-to-one sessions were more successful, several others (e.g. Pillay; Duke; Chrvala)
--

	have shown that group and one-to-one have shown similar outcomes.  • The theoretical orientation was not fully discussed for T2D. Only the specific theories used in ACS were noted. For example, empowerment and self-determination are common theories and were not discussed. Were behavioral strategies used for either the ACS or T2D identified (e.g., self-determined goal setting)? • It seems that on page 31, the “mortality and MI” statement should include short and long-term complications. • Reference 76 could be replaced with the more recent position statement by Powers (Powers M 2015, et al, Diabetes Care).
--	--

REVIEWER	Dr Hayley McBain City, University of London, UK
REVIEW RETURNED	23-May-2017

GENERAL COMMENTS	This is a well written and considered umbrella review, which I recommend for publication subject to some minor amendments:  1. Have you considered the overlap between systematic reviews? This could have considerable impact on the findings of this umbrella review. I would suggest you use the Corrected Cover Area (CCA) method (Pieper et al 2014) to calculate this and then consider how it impacts on your findings. 2. Page 10 - there is a difference in the number of articles that did not meet the inclusion criteria. Is this 371 or 372? 3. Page 10 - If review quality was used to remove studies please add this to the inclusion/exclusion criteria. 4. Page 10 - Do the figures for the total number of participants, take into consideration duplicate primary research papers across reviews? 5. Page 17, line 16 - How did you define behavioral change as opposed to self-management, was this pre-defined (if so please provide your definitions in the methods) or did you use the definitions provided by the authors of the reviews? 6. Discussion - I would ask you to consider whether combining ACS and DM together in each of the sections would be more useful, enabling comparisons between the two conditions. 7. Table 5 - it is unclear why the recommendations for health programs for both ACS and DM have been influenced more by the findings from the ACS studies. Please clarify.
---

VERSION 1 – AUTHOR RESPONSE

Reviewer: 1 Reviewer Name: Professor Martha Funnell	Authors' Responses
This well-written manuscript addresses an area that is of great interest to professionals who provide care and education for people with ACS and T2D. The umbrella methodology is clearly described, innovative and comprehensive. This manuscript provides a systematic approach to this topic that could be valuable to clinicians and researchers.	Thank you for this comment.

It was not clear that the practical differences between T2D and ACS education were appreciated and how these differences influence the study methods. For example, very few people with T2D are diagnosed in the hospital or are referred to education after “discharge”. The vast majority of T2D programs are offered in outpatient facilities. In addition, many of the recent studies were designed to meet national or international education standards (e.g., American Diabetes Assn, International Diabetes Assn) and payer requirements. This needs to be referred to understand the context of the reviews.	We have revised this in the text. We have inserted the following statement within the “Introduction” section (Page 6, line 1-6): “Most diabetes education is provided through programs within outpatient services or physicians’ practices.¹² Many recent education programs have been designed to meet national or international education standards,¹³⁻¹⁵ with diabetes education being individualized to consider patients’ existing needs and health conditions.¹⁶ Patients with T2DM have reported feelings of hopelessness and fatigue with low levels of self-efficacy, after experiencing an acute coronary episode.¹⁷”
On page 7 in the Intervention Type section: List the health professionals in the order of frequency of delivery. Nurses (including diabetes nurse educators) are later noted as most often providers, not community healthcare workers, doctors etc. as stated.	We have revised this in the “Intervention Types” section (Page 8 and 22) as follows: Page 8: “The interventions were delivered by nurses (including diabetes nurse educators), physicians, community health care workers, dietitians, lay people, rehabilitation therapists, or multidisciplinary teams.” Page 22: “Nurses (including diabetes nurse educators), community workers, dieticians and....”
Please clarify that “no evidence” includes only measured outcomes. Also, define “self-management in 20 reviews” in T2D. Could these be considered standardized diabetes education programs that meet national or international guidelines?	We confirm that only measured outcomes were included. We have added a definition for “self-management educational interventions” according to National Standards for Diabetes Self-Management Education and Support in the page 15. “(activities that promote or maintain the behaviors to manage T2DM often based on the National Standards for Diabetes Self-Management Education¹³)”
Pages 20 and 21: Please state how effectiveness is defined in “appears to ineffective for HA1C”. In addition, improving adherence to medical treatment is very vague. Do you mean medication taking?	We have changed “they appear ineffective for HbA1c control” to “but they were ineffective for reductions in HbA1c scores^{71 72}.” in the page 16. We have improved this section as per below (page 15 and 17): Page 15: “improving compliance in taking medication interventions (e.g., promoting oral

The word “diabetic” is not used to refer to either people with diabetes or outcomes. Several organizations (included Australia Diabetes Educators) have recent position statements on this issue.	hypoglycemic adherence),” Page 17: “Improving Adherence to Medication Regimes The statements are based on our synthesis of results from three publications.^{57 79 80} There is some evidence of the effectiveness of improving adherence to taking medications for HbA1C control including oral hypoglycemic agents.” We have changed “diabetic complications” and “diabetic control outcomes” to “diabetes complications” and “diabetes control outcomes” in the revised manuscript (page 16 and 21).
Were more recent psychosocial measures than of quality of life, such as disease-related distress assessed in the articles reviewed?	There is no difference between psychosocial measures (such as disease-related distress, depression, anxiety and psychological wellbeing) and quality of life (QoL) in the included articles related ACS and T2DM. In ACS reviews, 2 reviews (published at 2013 and 2014) measured psychosocial measures, one review (published at 2013) measured QoL, and another one review (published at 2010) measured both outcomes (QoL and psychosocial measures). In diabetes reviews, 7 reviews (published at 2016, 2014, 2013, 2012, 2009, 2008 and 2005) measured psychosocial outcomes. Five reviews (published at 2014, 2013, 2010, 2008 and 2007) measured QoL, two reviews (published at 2014 and 2001) measured both outcomes (QOL and psychosocial measures). We did not include any statements based on these findings.
Although one study for T2D reported that one-to-one sessions were more successful, several others (e.g. Pillay; Duke; Chrvala) have shown that group and one-to-one have shown similar outcomes.	We have inserted the following sentence to address this point. (Page 22). “However, one systematic review reported that individual and group patient education, demonstrated similar outcomes among T2DM patients.^{46”}
The theoretical orientation was not fully discussed for T2D. Only the specific theories used in ACS were noted. For example, empowerment and self-determination are common theories and were not discussed.	We have revised this in the text (Page 19, the table 5). Page 19: “13 of the 36 reviews (36.11%) related to T2DM reported the theoretical approach used in their included studies. The most common theories were SCT (including

Were behavioral strategies used for either the ACS or T2D identified (e.g., self-determined goal setting)?	self-efficacy theory), empowerment theories (eg., Empowerment Behavior Change Model, Self-determination and Autonomy Motivation Theory, Middle-range Theory of Community Empowerment) and TTM.” The behavioral strategies such as goal setting by patients or health professionals or mutually-agreed goal, used for either the ACS or T2D were identified (Page 23, 24 and the table in page 29). Page 19: “Three reviews^{31 41 44} noted that some included studies used behavioural strategies such as goal setting. These strategies were found to be beneficial for patients with coronary heart disease.” Page 20: “Fourteen reviews^{30 33 40 46 52 57 60 63 64 67 68 73 75 77} reported that goal setting was conducted in the included studies. Goal setting by patients, health professionals or mutually-agreed goals were linked to improved patient outcomes.”
It seems that on page 31, the “mortality and MI” statement should include short and long-term complications.	“Short and long-term complications” is inserted in “Conclusion” section (Page 27). “The effectiveness of these programs was based on HbA1C levels, knowledge, psychosocial outcomes, readmission rates, and smoking status rather than clear evidence of reduced mortality, MI, or short and long-term complications.”
Reference 76 could be replaced with the more recent position statement by Powers (Powers M 2015, et al, Diabetes Care).	We have revised this as requested (page 20). “^{75 83} Theories could help to specify the key target health behaviors and behavioral change techniques required to generate the desired outcomes.”
Reviewer: 2 Reviewer Name: Dr Hayley McBain	
This is a well written and considered umbrella review, which I recommend for publication subject to some minor amendments:	Thank you.

1. Have you considered the overlap between systematic reviews? This could have considerable impact on the findings of this umbrella review. I would suggest you use the Corrected Cover Area (CCA) method (Pieper et al 2014) to calculate this and then consider how it impacts on your findings.	We have revised this in the text (Page 9-11, 26) and Appendix 2 and 3. The overall overlap of studies among reviews or meta-analyses related ACS and T2DM was slight, CCA of 2.6% and 2.1%, respectively. We have included the following section on page 9 providing an explanation of the statistic: “This umbrella review calculated the Corrected Covered Area (CCA) (Appendix 2, 3). The CCA statistic as a measure of overlap of trials (the repeated inclusion of the same trial in subsequent systematic reviews included in an umbrella systematic review). A detailed description of the calculation is provided by the authors who note slight CCA as 0-5%, moderate CCA as 6-10%, high CCA as 11-15% and very high CCA is more than 15%.²⁵ The lower the CCA the lower the likelihood of overlap of trials included in the umbrella review.” Page 10-11: “The overlap of the trials included in the 15 reviews and meta-analyses related to ACS was slight (CCA = 2.6%). For the 36 systematic reviews relating to T2DM, the overlap of trials within these 35 reviews and meta-analyses (one review⁴⁷ did not report the included studies) was slight (CCA = 2.1%).” Appendix 2 and 3 provide supplementary data demonstrating the calculation, using an index case and repeated cases for both ACS and T2DM systematic reviews. Also inserted Page 26 the following statement has been added: “In addition, 240 studies were included more than once in the included reviews and meta-analyses. However, the overall overlap of studies among reviews and meta analyses related ACS and T2DM was slight, CCA of 2.6% and 2.1%, respectively.^{25”}
2. Page 10 - there is a difference in the number of articles that did not meet the inclusion criteria. Is this 371 or 372?	We have revised this in the “Characteristics of Included reviews” section (page 10). “The database search yielded 692 publications, with removal of 197 duplicates and 371 articles that did not meet the inclusion criteria,....”

3. Page 10 - If review quality was used to remove studies please add this to the inclusion/exclusion criteria.	We have included the sentence below in the "Assessment of Methodological Quality" section (Page 9, line 2-3). "The low quality reviews (AMSTAR scale: 0-3) were excluded in this umbrella review."
4. Page 10 - Do the figures for the total number of participants, take into consideration duplicate primary research papers across reviews?	Further details highlight this important issue have been inserted under "Characteristics of Included reviews" section (Page 11). "The average sample size of included articles was 8,161 (range, 536-68,556) participants, however, 63 studies related to ACS and 177 studies related to T2DM, were included in more than one systematic review or meta-analysis (see Appendix 2 and 3 and CCA statistics). The sample of these studies would therefore be included more than once."
5. Page 17, line 16 - How did you define behavioral change as opposed to self-management, was this pre-defined (if so please provide your definitions in the methods) or did you use the definitions provided by the authors of the reviews?	The definitions of education content were different in the included reviews. In order to make the definitions of education contents more specific and clear, this umbrella review used the definitions that described in the included reviews and did not pre-defined the topics of education contents.
6. Discussion - I would ask you to consider whether combining ACS and DM together in each of the sections would be more useful, enabling comparisons between the two conditions.	We have revised the "Discussion" section to reflect this in the revised manuscript and combined ACS and T2DM together in each of the sections (page 18-24).
7. Table 5 - it is unclear why the recommendations for health programs for both ACS and DM have been influenced more by the findings from the ACS studies. Please clarify.	We have inserted the following statement (Page24): "The acute life-threatening nature of ACS requires that increased emphasis should be placed on cardiovascular risk factors in any combined education program. Both ACS and T2DM have common lifestyle factors such as inactivity and high fat diet requiring modifications."

VERSION 2 – REVIEW

REVIEWER	Martha Funnell University of Michigan Ann Arbor, MI USA
REVIEW RETURNED	27-Jun-2017

GENERAL COMMENTS	Thank you for your responsiveness to the comments.
--

	Along with diabetic, the words control, compliance and adherence are considered pejorative.
--	---

VERSION 2 – AUTHOR RESPONSE

Reviewer: 1 Reviewer Name: Professor Martha Funnell	Authors' Responses
Thank you for your responsiveness to the comments. Along with diabetic, the words control, compliance and adherence are considered pejorative.	Thank you for this comment. We have revised this in the text and have been highlighted in red (page 5, 13-27, and 58). Page 5: “medication taking”. Page 13: “glycemic regulation in 16 reviews (44.45%)” Page 15: “improving the uptake and maintenance of medication regimes (e.g., promoting the use of oral hypoglycemic medications)”. Page 16: “HbA1c reduction”; “on the management of glycemia, weight reduction, and some diabetes management outcomes”; and “body weight”. Page 17: “Uptake and maintenance of medication regimes”; “of increased uptake and maintenance of medication regimes for taking medications for HbA1C regulation including oral hypoglycemic agents.” “HbA1c level,”; “body weight”; “weight management” and “HbA1c regulation”. Page 21: “glycemic regulation”; and “impact on the management of glycemia”. Page 22: “blood glucose reduction and knowledge levels in”; “blood glucose regulation” and “metabolic management”. Page 24: “blood glucose reduction and knowledge levels among” and “glucose reduction”. Page 27: “glycemic level”. Page 58: “GC= glycemic regulation;”.